# Molecular basis of force-from-lipids gating in the mechanosensitive channel MscS

**Bharat Reddy[1], Navid Bavi[1], Allen Lu[1], Yeonwoo Park[2], Eduardo Perozo[1,3]***

[1]Department of Biochemistry and Molecular Biology, The University of Chicago, Chicago, United States; [2]Department of Ecology and Evolution, The University of Chicago, Chicago, United States; [3]Institute for Biophysical Dynamics, The University of Chicago, Chicago, United States

**Abstract** Prokaryotic mechanosensitive (MS) channels open by sensing the physical state of the membrane. As such, lipid-protein interactions represent the defining molecular process underlying mechanotransduction. Here, we describe cryo-electron microscopy (cryo-EM) structures of the *E. coli* small-conductance mechanosensitive channel (MscS) in nanodiscs (ND). They reveal a novel membrane-anchoring fold that plays a significant role in channel activation and establish a new location for the lipid bilayer, shifted ~14 Å from previous consensus placements. Two types of lipid densities are explicitly observed. A phospholipid that 'hooks' the top of each TM2-TM3 hairpin and likely plays a role in force sensing, and a bundle of acyl chains occluding the permeation path above the L105 cuff. These observations reshape our understanding of force-from-lipids gating in MscS and highlight the key role of allosteric interactions between TM segments and phospholipids bound to key dynamic components of the channel.

## Introduction

In principle, all molecules are mechanosensitive. Remarkably, some have evolved as force transducers, where they participate in a variety of fundamental biological functions, including turgor control in plants, development and morphogenesis, touch, hearing, proprioception, as well as osmoregulation in bacteria (*Haswell et al., 2011*; *Katta et al., 2015*; *Kung, 2005*; *Ladoux and Mège, 2017*; *Murthy et al., 2017*). Many of these functions are driven by the activity of mechanosensitive channels, switches that couple force sensing with the electrical activity of cells (*Cox et al., 2018*; *Naismith and Booth, 2012*; *Perozo, 2006*; *Sukharev and Corey, 2004*). When the membrane is stretched, the resulting change in the trans-bilayer pressure profile will drive the conformational equilibrium of membrane proteins (*Cantor, 1999*; *Gullingsrud and Schulten, 2004*). To explain this phenomenon, several physical properties of the lipid bilayer have been considered (*Perozo, 2006*). Nevertheless, understanding the nature of the lipid forces that drive MS channel gating continues to be one of the fundamental questions in biological mechanotransduction.

In prokaryotes, MscL and MscS are the foundational members of two structurally distinct classes of MS channels (*Cox et al., 2018*; *Kung et al., 2010*; *Naismith and Booth, 2012*). Sensitive to tension changes in the plane of the bilayer, they play a key role in the response to osmotic challenges and remain a *de facto* standard in the search for the molecular principles underlying membrane force transduction. The *E. coli* MscS crystal structure in detergent (*Bass et al., 2002*; *Steinbacher et al., 2007*) revealed a homoheptamer with three TM segments (TM1, TM2 and TM3a/b) and a large cytoplasmic C-terminal domain cradling a water-filled internal cavity. The narrowest region along the permeation path is located at L105 in TM3a, and while its van der Waals diameter can be larger than 7 Å, the current consensus is that the original crystal structure represents a non-conductive conformation (*Anishkin et al., 2010*). However, whether it corresponds to a closed or an inactivated state has been a matter of discussion (*Anishkin and Sukharev, 2004*; *Anishkin et al., 2008a*;

*For correspondence:
eperozo@uchicago.edu

**Competing interests:** The authors declare that no competing interests exist.

*Anishkin et al., 2008b*). Structures believed to represent a conducting or expanded state have been obtained for the gain of function (GOF) mutant A106V (*Wang et al., 2008*) and in DDM-solubilized wt-MscS (*Lai et al., 2013*). A comparison with wt-MscS revealed that transition to this expanded state appears to be associated with rearrangements in the packing interface between TM3 helices. These crystallographically-derived conformational changes (in detergent) are in general agreement with lower resolution analyses of amphiphile-driven gating of MscS in a lipid bilayer (*Vasquez et al., 2008*).

Given the existing set of MscS structures, a few plausible mechanisms have been proposed to describe how bilayer forces lead to channel opening. Early hypotheses suggested that membrane lateral forces at the aqueous interface (*Nomura et al., 2006*) disrupt the hydrophobic interactions between TM1-TM2 hairpin, straightening the tilt angle of TM3a and TM3b and opening the channel (*Anishkin et al., 2008b*; *Vasquez et al., 2008*). Alternatively, at rest, internal elastic strain energy might simply be countered by the bilayer pressure profile, keeping the channel closed (*Anishkin et al., 2008b*; *Malcolm et al., 2015*). More recently, a unique mechanism has been proposed based on the idea that bilayer lipids can drive MscS conformation by acting as ligands while they freely exchange between the membrane and hydrophobic pockets between TM2 and TM3a (*Pliotas et al., 2015*). At rest, these pockets are saturated with phospholipids, preventing structural rearrangements in TM3a. But as tension is applied, lipids diffuse away, leading to TM3a movements and opening the channel. This model requires not only that the TM2/TM3a cavity must be located at the membrane/water interface but it also takes advantage of the presumed membrane deformations induced by the interaction of the 'angled' TM1-TM2 seen in the nonconductive MscS crystal structures (*Phillips et al., 2009*).

Understanding the fundamental role played by bilayer forces in MS channel gating requires the elucidation of channel structures in a lipid bilayer environment. Yet, until recently (*Rasmussen et al., 2019*), all available high-resolution structures have been determined in detergent micelles, either by crystallographic (*Bass et al., 2002*; *Dong et al., 2015*; *Lai et al., 2013*; *Pliotas et al., 2015*; *Steinbacher et al., 2007*; *Wang et al., 2008*) or single particle cryo-EM methods (*Guo and MacKinnon, 2017*; *Saotome et al., 2018*; *Zhao et al., 2018*). Solved in the absence of the lipid bilayer, these structures have been essential in understanding the structural basis of MS channel function, but cannot, on their own, elucidate the central role that lipid-protein interactions play in force-from-lipid mechanotransduction. However, in a recent EM structure of MscS in a nanodisc (*Rasmussen et al., 2019*), lipids were observed in the TM2-TM3 hairpin cavity, two lipids parallel to TM3b, and a putative lipid density was reported along the permeation pathway. Though not explicitly stated, this model also suggests a shift in the membrane footprint. Independently, we have solved the nanodisc-reconstituted MscS (MscS-ND) by single particle cryo-electron microscopy (cryoEM) under a variety of lipid compositions and protein constructs (*Figure 1—figure supplement 1*) as well as a DDM detergent structure. These structures, together with electrophysological and computational data highlight new membrane-interacting regions at MscS N-terminal end and define structurally bound lipids with potentially important roles in mechanotransduction, gating, and permeation.

## Results

### Structure of membrane-embedded MscS

We first determined the structure of nanodisc-reconstituted MscS (PC:PG, 4:1) with a histidine tag at its N-terminal end (6xHis-MscS-ND) to a resolution of 3.1 Å (*Figure 1—figure supplement 2*). While the EM structure (*Figure 1A*) recapitulates some of the major characteristic seen in the MscS crystal structure (*Bass et al., 2002*; *Steinbacher et al., 2007*), the channel shows a slightly different angle for the TM1-TM2 hairpin and displays additional density towards its periplasmic face. This is a consequence of a newly resolved N-terminal domain and additional ~3 turns of helix that further extends TM1 in MscS-ND (*Figure 1A*, right). This new density is also present in the absence of imposed symmetry (*Figure 1—figure supplement 3*). This domain displays some anisotropic behavior, and together with the lower end of the TM1-TM2 hairpin, they represent the most flexible regions of the channel (*Figure 1—figure supplement 2*).

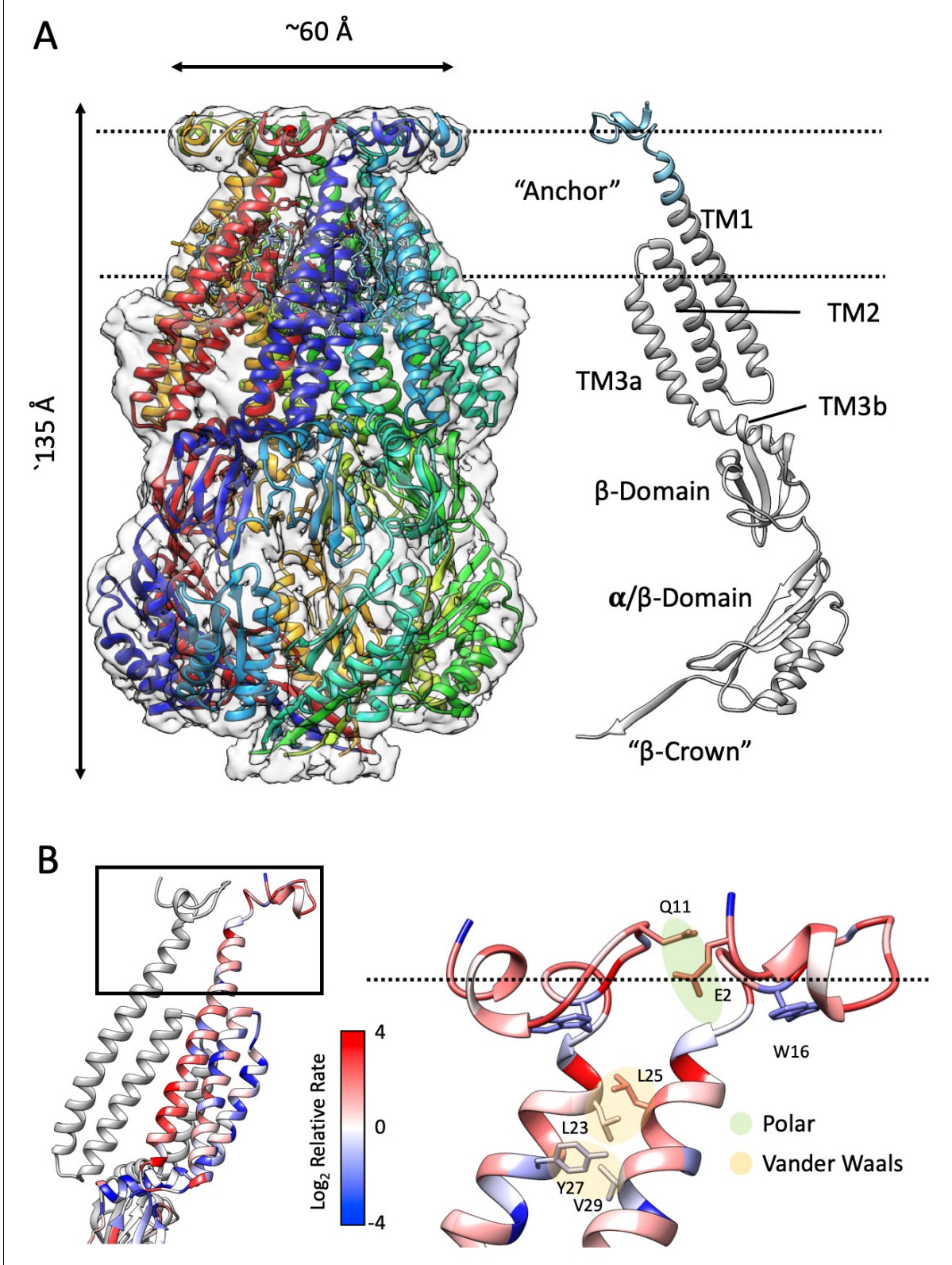

**Figure 1.** Structure of membrane-embedded MscS (MscS-ND) and its anchor domain. (**A**) Left, the 3.1 Å resolution structure of the nanodisc-reconstituted (E3D1) MscS heptamer, shown in cartoon representation. Each subunit is shown in a different color. Bound lipids are shown as stick representation. The transparent EM density is shown overlapped to the cartoon of the protein. Right, cartoon diagram showing the MscS monomer. Colored grey are regions of the channel resolved in the crystal structure (2OAU), regions newly resolved in the MscS-ND structure are shown in cyan.

*Figure 1 continued on next page*

*Figure 1 continued*

The putative location of the lipid bilayer is shown as a pair of dashed lines. (B) Residue conservation and inter-subunit interactions stabilizing the anchor domain. Highly conserved sites are shown in blue, variable sites in red. Shown in sticks and balls representation are residues participating in inter-subunit interactions, either polar in nature (E2–Q11) or hydrophobic (van der Waals) packing (L23–L24, Y27–V29). On the left, a cartoon representation of the TM segments in two adjacent subunits, where the top box indicates the location of the diagram on the right.

The online version of this article includes the following figure supplement(s) for figure 1:

**Figure supplement 1.** MscS Constructs, Nanodisc Composition, and Purification MscS Nanodiscs.
**Figure supplement 2.** Overview of MscS ND Density Refinement Workflow.
**Figure supplement 3.** Symmetry Free Processing of MscS ND.
**Figure supplement 4.** Bioinformatics of the MscS N-terminal Anchor Domain.
**Figure supplement 5.** MscS-ND vs.other Models.

The N-terminal domain sits atop TM1, forming a returning loop that projects away from the seven-fold symmetry axis and lines the periphery of an N-terminal ring at the periplasmic face of the channel (*Figure 1B*). Despite the overall lower resolution in the region, we were able to fit a Cα backbone and buildout sidechains using the existing density information and molecular dynamic flexible fitting (*Croll, 2018*). The N-terminal ring is potentially stabilized by both polar (E2-Q11) and hydrophobic (van der Waals) (L23-L24, Y27-V29) interacting pairs at the inter-subunit interface (*Figure 1B*). As will be described below, a key finding of the MscS-ND structure is that this newly resolved N-terminal domain is membrane-embedded and seems to dominate its interactions with the outer leaflet of the bilayer. Comparing the sequences of MscS homologs revealed that this particular structural motif is largely present in *Enterobacteriales* (*Figure 1—figure supplement 4*). *Figure 1B* shows the relative rate of evolution of individual residues mapped on the newly resolved domain. Conserved residues (in blue) are predominantly found at the subunit interface, while W16 sits deep in a pocket at the bottom of the N-terminal domain. Indeed, tryptophan residues have been shown to be enriched at the membrane interface and contribute about ~4 kcal/mol as 'anchors' of TM segments in membranes (*de Jesus and Allen, 2013*). Given its membrane placement and location of the conserved W16, we named the MscS N- terminal the 'anchor' domain.

Strikingly, other than the upper third of TM1 and anchor domain the structure of MscS-ND is rather similar to the MscS crystal structure in detergent (2OAU) (*Steinbacher et al., 2007*) (~1.4 Cα RMSD, *Figure 1—figure supplement 5*). Previous attempts to reconfigure the interactions between TM2 and TM3 to compensate for perceived low inter-helical packing (*Anishkin et al., 2008a*; *Vásquez et al., 2008*) are not supported by the present data. In fact, we reason that the present EM structure represents the physiological closed state: The channel is at rest, embedded in a lipid bilayer and ostensibly, in the absence of any applied tension. Furthermore, as MscS-ND includes a 6xHis tag at the N-terminus (with a 10 residue linker), clear density corresponding to the oligomeric assembly of individual Hisx6 tags is observed as a 'crown' on top of MscS (*Figure 2A* left, *Figure 3*, left and *Figure 4—figure supplement 1*). This crown leads to the formation of a strong stabilizing force, which precludes N-terminal movement in the closed state. Interestingly, the quality of the density of the anchor domain is significantly degraded in the absence of the N-terminal Hisx6 tag (*Figure 3*, left-center), presumably due to an increase in local dynamics in the absence of the stabilizing His-tag. Besides being a fortuitous result, the N-terminal Hisx6 serves as a useful probe of MscS conformational changes. 6xHis-MscS-ND appears insensitive to mechanical stimulation in HEK293 cells patch-clamp experiments under high-speed pressure clamp conditions (*Figure 4—figure supplement 1C*). This functional inhibition is fully relieved upon in situ thrombin treatment (*Figure 4—figure supplement 1D*), strongly suggesting that the anchor domain must move away from its current position upon channel opening, in a way consistent with the MscS expanded conformation (*Lai et al., 2013*; *Wang et al., 2008*).

To evaluate the role of the anchor domain on MscS function, we designed a construct lacking the first 26 residues of the channel: Δ2–26 MscS ('Cryst'), physically recapitulating the resolved regions crystal structure (2OAU) model, which has been the basis of many past experimental and computational insights (*Cox et al., 2018*; *Naismith and Booth, 2012*; *Perozo, 2006*; *Sukharev and Corey, 2004*). This construct displays a severe loss of function (LOF) phenotype and is unable to elicit any mechanically activated currents in either HEK293 cells patch clamp experiments (*Figure 4A*) or in downshock assays (*Figure 4B*). To further evaluate the functional role of the anchor domain, we

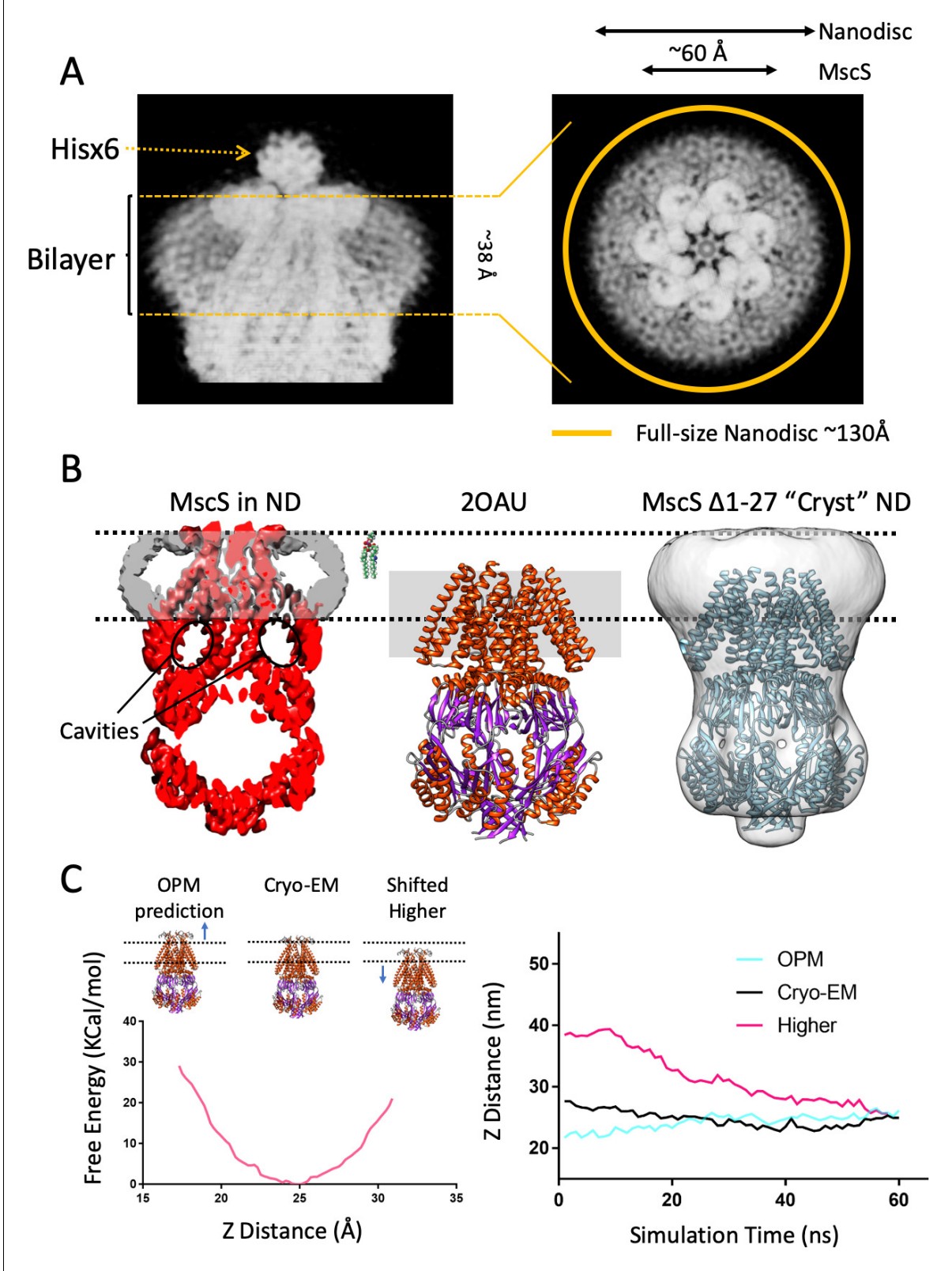

**Figure 2.** A new lipid-protein interface for membrane-embedded MscS. (**A**) Close-up of MscS-ND EM density (in Chimera's 'solid' representation). Left, Side view. The location of the bilayer in the nanodisc in indicated by dashed yellow likes (approximately 38 Å in diameter). Density for the putative heptameric histidine tag complex is shown by a dotted arrow. Right, Top view. The yellow circle represents the putative average size of the E3D1 nanodisc (~130 Å) in relation to the density, which points to a partial averaging of the density likely due to MscS lateral mobility. (**B**) Comparison
*Figure 2 continued on next page*

*Figure 2 continued*

between the location of the membrane interface in MscS-ND, the FC14 crystal structure (2OAU) and the 'Cryst' deletion construct. Black dashed lines depict the limits of the lipid bilayer based on the nanodisc EM density. Left, EM density for the protein (red) and the nanodisc (grey) for MscS-ND, the black ovals highlight the fact that the prominent cavities formed between the TM1-TM2 hairpin and TM3 are fully located outside the membrane. Center, relative positioning of 2OAU based on a rigid fit of the structure onto MscS-ND EM density. The gray rectangle in the background represents the previous consensus membrane location. Right, the low-resolution cryoEM structure of MscS Δ2–27 ('Cryst',~20 Å) shows an overall architecture for the nanodisc-embedded channel. In spite of the N-terminal deletion, the nanodisc is located at the same position as in MscS-ND. (C) Probing the energetics of the membrane interface. A Potential of mean force (PMF) calculation was carried out by relocating a lipid bilayer from a coordinate origin (0 Å) predicted by the CHARM-GUI server (*Jo et al., 2008*) and moved up to 16 Å (the thickness of a lipid monolayer) along the Z-axis coordinate (see *Figure 2—figure supplement 2*). Left, free energy as a function of Z-axis displacement. A global minima was found at ~25 Å (~8 Å above the prediction) and the free energy increases exponentially beyond this point. The energy minima coincides with the location if the interface as defined by the EM density of MscS-ND. Right, evolution of MD simulation starting at three membrane interface locations: predicted by CHARM-GUI (0 Å, red trace), at the cryo-EM density (+ 8 Å, black trace) and a further +16 Å (Higher placement, blue trace). After ~60 ns simulation all membrane interfaces converge to that defined by the cryo-EM density.

The online version of this article includes the following figure supplement(s) for figure 2:

**Figure supplement 1.** MscS bilayer footprint is compatible with bilayer predictions and surface charge distribution.
**Figure supplement 2.** Details of MD simulations and PMF calculated from umbrella sampling for determining the optimum position of MscS with respect to the bilayer.
**Figure supplement 3.** Geometrical properties of MscS embedded in a lipid bilayer for PMF calculations.
**Figure supplement 4.** Cartoon representation of concentric areas and associated curvatures around membrane-embedded MscS.

---

carried out an alanine scan at residues 2–30 for in vivo analysis of MS channel activity (*Figure 4B*). The effect of alanine substitutions was dramatic. About 70% of the mutants displayed LOF phenotypes, 8 of those severe (E2A, V6A, S9A, G12A, N20A, Q21A, L24A, L25A). Some of these severe LOF mutations might be potential GOF mutations due to their survival being lower than the negative

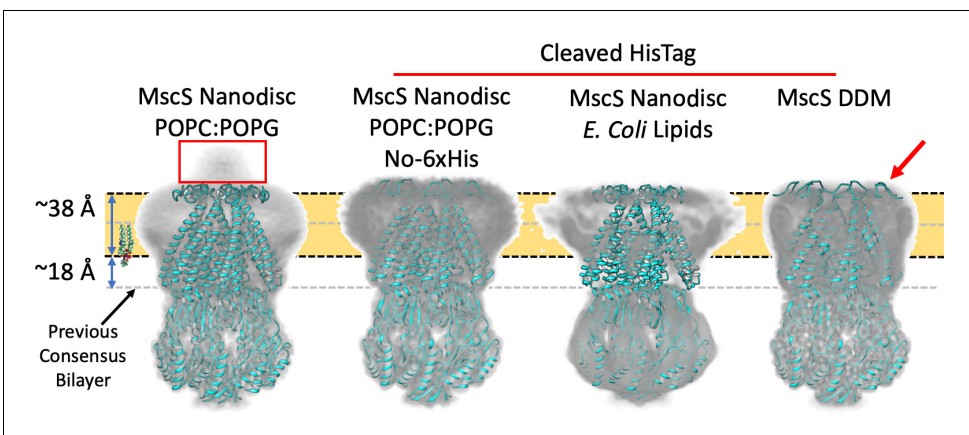

**Figure 3.** Membrane interface location in nanodisc-reconstituted MscS is independent of lipid composition. CryoEM structures for three additional lipid reconstitution/detergent conditions show a common membrane interface. Four independently determined structures are shown: Left, MscS-ND (in POPC:POPG 4:1) determined at 3.1 Å. This structure includes the N-terminal 6xHis (red rectangle). Center left, MscS-ND (in POPC:POPG 4:1) after thrombin proteolysis of the N-terminal 6xHis, determined at 4.1 Å. Center right, MscS-ND (in *E. coli* lipids) after thrombin proteolysis of the N-terminal 6xHis, determined at ~10 Å. Right, DDM-solubilized MscS after thrombin proteolysis of the N-terminal 6xHis, determined at 3.4 Å. In all cases, EM density is shown as Chimera's 'solid' representation with the protein depicted in ribbon representation (cyan). The calculated location of the membrane is shown as a yellow slab with black dash lines, while the previous membrane interface consensus location is represented by the grey dashed lines. The red arror points to a partially unfolded region of the N-terminal loop region.

The online version of this article includes the following figure supplement(s) for figure 3:

**Figure supplement 1.** Overview of MscS ND No His-Tag Density Refinement Workflow.
**Figure supplement 2.** Overview of MscS DDM Density Refinement.
**Figure supplement 3.** Model Fit to Density.

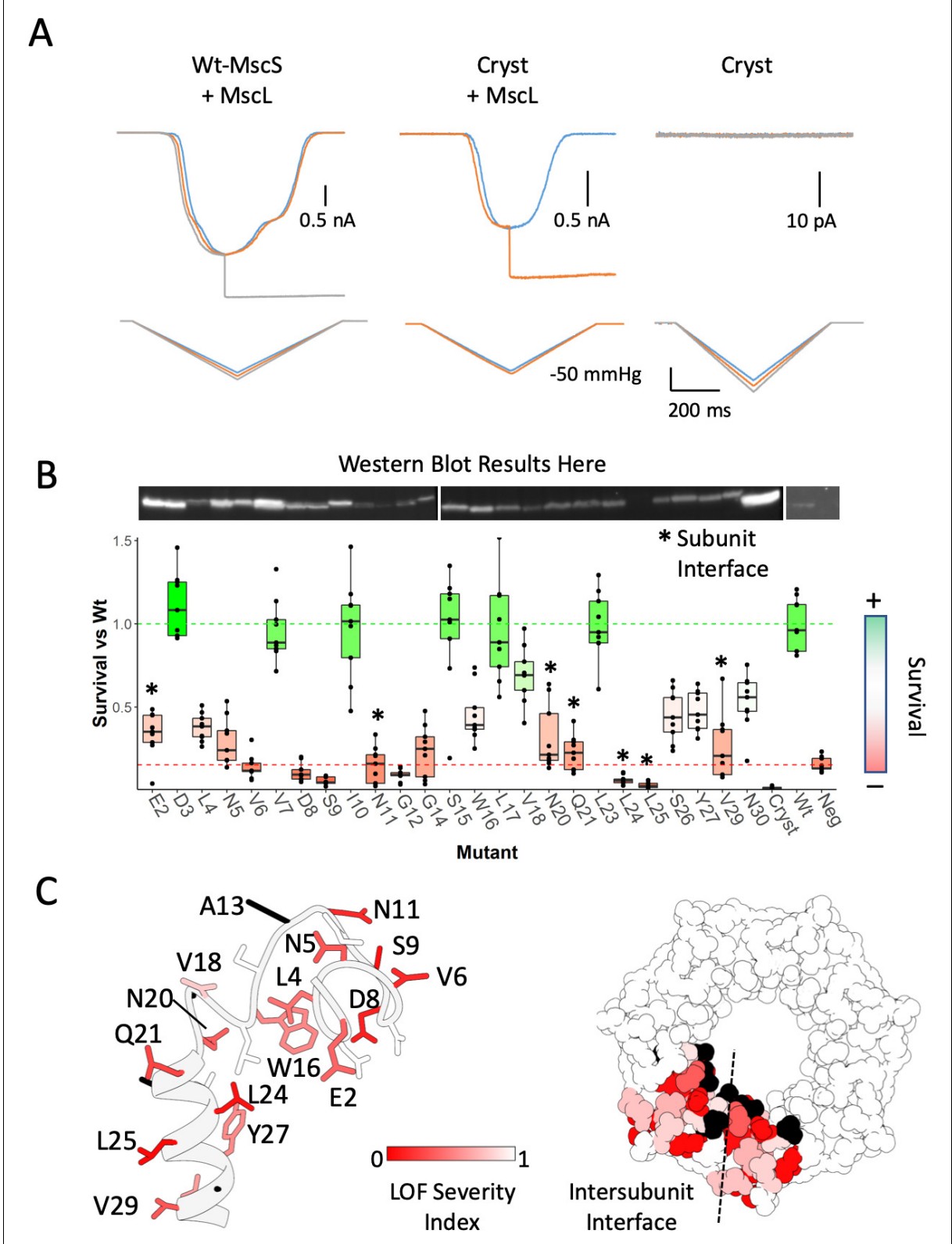

**Figure 4.** Functional significance of the anchor domain. (**A**) Functional consequences of deleting the anchor domain (MscS Δ1–27). High-speed pressure clamp recordings of co-expressed MscS and MscL in HEK296 cells under voltage clamp conditions. A family of macroscopic currents elicited by pressure ramps are shown for co-expressed wt-MscL and wt-MscS (left traces), wt-MscL and Cryst (center traces) or Cryst alone (right traces). (**B**) Osmotic downshock assays of N-terminal alanine scan mutants. Box plots are shown for nine independent experiments, where the central bar

*Figure 4 continued on next page*

*Figure 4 continued*

represents the median, the box 25%−75% quartiles and the individual data are depicted as black dots. Data were normalized to the wild-type behavior and the dotted lines represent the mean survival for the negative control (red, empty vector) and wt-MscS (green). Asterisks indicate residues located at the subunit interface. (C) Functional complementation of alanine scan mutants mapped onto the anchor domain structure. Left, side view of the anchor domain monomer. Right, top view of the anchor domain as a heptamer with data mapped on two adjacent subunits. The residues are colored by surivial relative to Wt (LOF Severity index) where 0 corresponds to no survival compared and one is equal or greater survival than wild-type. Black residues are wild-type alanines (not evaluated). The dashed line indicates the location of the subunit interface.

The online version of this article includes the following figure supplement(s) for figure 4:

**Figure supplement 1.** Functional consequences of an N-terminal 6xHis-Tag.
**Figure supplement 2.** Osmotic downshock assays of N-terminal Cystine Scan Mutants.

control of no MscS. A severe MscS GOF mutant will have dramatic effects on the survival of any cell, but final demonstration requires electrophysiological data under pressure clamp conditions. Remarkably, mapping the positions of the alanine LOF mutants (*Figure 4C*) show that they are located almost exclusively at the membrane or inter-subunit contact interfaces (*Figure 4C*, dotted line). These results not only highlight the importance of subunit-subunit interactions for MscS function, but point to a role of the anchor domain in MscS mechanotransduction. In fact, an earlier cysteine scan of the same region (*Figure 4—figure supplement 2*) (*Vásquez et al., 2008*), shows that almost a third of the mutants display a noticeable LOF phenotype (V7C, N11C, A13C, V18C, and L25C), two were severe (V6C and A19C) and most map to the subunit interface.

## Redefining MscS interaction with the lipid bilayer

Based on the MscS crystal structure (2AOU) (*Bass et al., 2002*; *Steinbacher et al., 2007*), a *de facto* consensus had been reached where the bilayer interacted with the channel at the level of the resolved regions of the TM segments (see *Booth and Blount, 2012*; *Cox et al., 2018*; *Edwards et al., 2004*; *Perozo, 2006*; *Pliotas and Naismith, 2017*; *Zhang et al., 2016*). In fact, early MD simulations of membrane-embedded MscS showed unexpected instabilities in the putative transmembrane segments, even under different force fields (*Anishkin et al., 2008a*; *Sotomayor and Schulten, 2004*; *Spronk et al., 2006*). We note, however, that a partial shift in membrane location had been previously suggested after modeling the N-terminus of the channel (*Anishkin et al., 2008a*). The present MscS-ND EM density now redefines the nature and extent of MscS interactions with the lipid bilayer. Remarkably, our data show that the disc-like region corresponding to the nanodisc lipid bilayer (*Figure 2A B* left) is upwards shifted ~14 Å when compared to the previous consensus placement (*Figure 2B*, center). This is also highlighted on side views from 2D classes (*Figure 1—figure supplement 2*).

Given this location, the membrane not only interacts closely with the N terminal domain (*Figures 1A,B,2A*) but a large portion of the cytoplasmic end of the TM1-TM2 hairpin and most of TM3a (including the location of the vapor lock) now lies outside of the membrane (and displays considerable degree of conformational heterogeneity (*Figure 1—figure supplement 2*, *Figure 3—figure supplement 1* and *Figure 3—figure supplement 2*). This arrangement is fully compatible with the distribution of MscS surface charged residues (*Figure 2—figure supplement 1A*). Furthermore, modern algorithmic predictions of membrane placement (*Lomize et al., 2012*; *Newport et al., 2019*) are in agreement with the present membrane location when the MscS-ND structure is evaluated (*Figure 2—figure supplement 1B,C*).

As a quantitative evaluation of the bilayer interaction energetics, we carried out a potential of mean force (PMF) calculation where MscS-ND was moved along the Z-axis in relation to a fixed bilayer (*Figure 2—figure supplement 2*). Calculation of the free energy as a function of linear displacement reveals a clear energy minima centered precisely at the location experimentally defined by the EM density (*Figure 2C*, left). Indeed, equilibration of three bilayer positions (upwards and downwards of MscS-ND placement) all converge to the EM density position within 60 ns (*Figure 2C*, right). When the free energy change based on our continuum calculation is compared to those obtained from PMF calculations, the contribution of hydrophobic mismatch dominates that of membrane curvature (*Figure 2—figure supplement 3*). Although the hydrophobic mismatch in our initial MD-PMF simulations was not systematically changed (the reaction coordinate was set to move the protein up/down across the bilayer thickness), the minima in the free energy obtained from our MD

simulation is matched with that obtained from our mean-field calculations (at the reaction coordinate for the PMF calculations has been defined as the distance between the Z coordinate of the center mass of phosphate molecules of the lipid bilayer and the Z coordinate of center mass of the pore-forming helices of MscS (i.e. residue 105 to 115), Z distance ~25 Å). Hence, our PMF calculations confirm the most energetically favorable position of MscS in the bilayer (at Z distance ~25 Å), otherwise the hydrophobic mismatch between the membrane and protein would have excess energetic costs (*Figure 2—figure supplement 3C*). These results must be evaluated by taking into formal consideration potential issues of force/tension bias once MscS transitions from one conformation to another within a nanodisc. However, we believe these to be relatively minor.

To expand on this result, we pursued additional MscS EM structures under various lipid/detergent conditions. Two specific questions were addressed: Is the location of the bilayer related to its lipid composition? What is the EM structure of MscS in DDM and what is its micelle placement? The structure of MscS-ND was determined in nanodiscs containing PC:PG 4:1, yet *E. coli* membranes are composed mostly of PE, PG, and cardiolipin (*Raetz and Dowhan, 1990*). *Figure 3* shows that the same bilayer placement is observed in the EM densities for two PC:PG MscS-ND structures (with and without N-terminal 6xHis, *Figure 3—figure supplements 1,3*) and in nanodiscs containing *E. coli* polar lipids (at 10 Å resolution), suggesting that MscS-bilayer placement is not affected by alternative lipid compositions. The EM density in DDM (at about 3.4 Å, *Figure 3—figure supplements 2,3*) appears to show a similar placement for the detergent micelle. However, the structure of the N-terminal domain has partially unraveled (*Figure 3*, red arrow), suggesting a rationale for why this region is unresolved under crystallographic conditions or in nanodiscs in the absence of the N-terminal 6xHis (*Rasmussen et al., 2019*).

## Lipids bound to dynamic regions of MscS

At the present resolution, the MscS-ND maps provide an excellent template to evaluate the nature and extent of the interactions between the channel and the lipid bilayer. Notably, we find lipid-like densities both, in a newly defined cavity between subunits and at the center of the permeation pathway (*Figure 5A*). Seven clearly defined phospholipids appear to 'hook' the top of each of the TM2-TM3 loops, a region that putatively displays large conformational rearrangements during gating (*Lai et al., 2013*; *Wang et al., 2008*). The hook lipids are firmly embedded by threading their head group through an inter-subunit opening formed by the extended TM1 and the top of the TM2-TM3 hairpin (*Figure 5—figure supplement 1A–B*) and facing the permeation path. At the present resolution, the nature of the hook lipids head group was not defined, but was ultimately modeled as PC due to the nanodisc composition. But given the *E. coli* membrane composition, it is likely that the hook lipids are PE or PG.

Given that hook lipids bind to a conformationally active region of MscS, we suggest they might participate in the transduction of bilayer forces that influence the conformation of the MscS gate (TM3a). Indeed, binding of the hook lipids is stabilized by R88 in one subunit and Y27 from the neighboring subunit (*Figure 5B*). Both residues are known to generate LOF phenotypes when mutated (*Figure 4D*; *Rasmussen et al., 2015*). Additional residues along TM1 and the TM2-TM3 linker participate via hydrophobic contacts (*Figure 5B*). A cluster of seven linear densities is also observed lining the patch of hydrophobic residues immediately above the narrowest region of the permeation pathway (L105) (*Figure 5—figure supplement 1C–E*). Although there is no clear indication of EM density associated with headgroups (whether it is PE or PG), we hypothesize these are either acyl chains from a fairly mobile (and not fully resolved) phospholipid or perhaps bound fatty acids trapped along the permeation pathway. Modeled as hexadecanes, the pore lipids are stabilized (likely weakly) via hydrophobic interactions along TM3 (*Figure 5C*). Though not currently resolved, a potential pore lipid head group might thread through an inter-subunit gap between the G104s in TM3a (*Figure 5—figure supplement 1F*). Interestingly, a tryptophan scan encompassing TM3a and the TM2-TM3 linker (*Rasmussen et al., 2015*) shows a remarkable correlation with LOF residues aligning at the putative hook lipid binding pocket and the pore lipids interaction surface (*Figure 5—figure supplement 2B*).

We believe the pore and hook lipids were carried over from the cell membrane during purification and reconstitution, as we find evidence of similarly placed density in our DDM MscS map (*Figure 5—figure supplement 3A*). Furthermore, an evaluation of the electron density map for the 2OAU crystal structure (*Steinbacher et al., 2007*) displays unassigned density that precisely

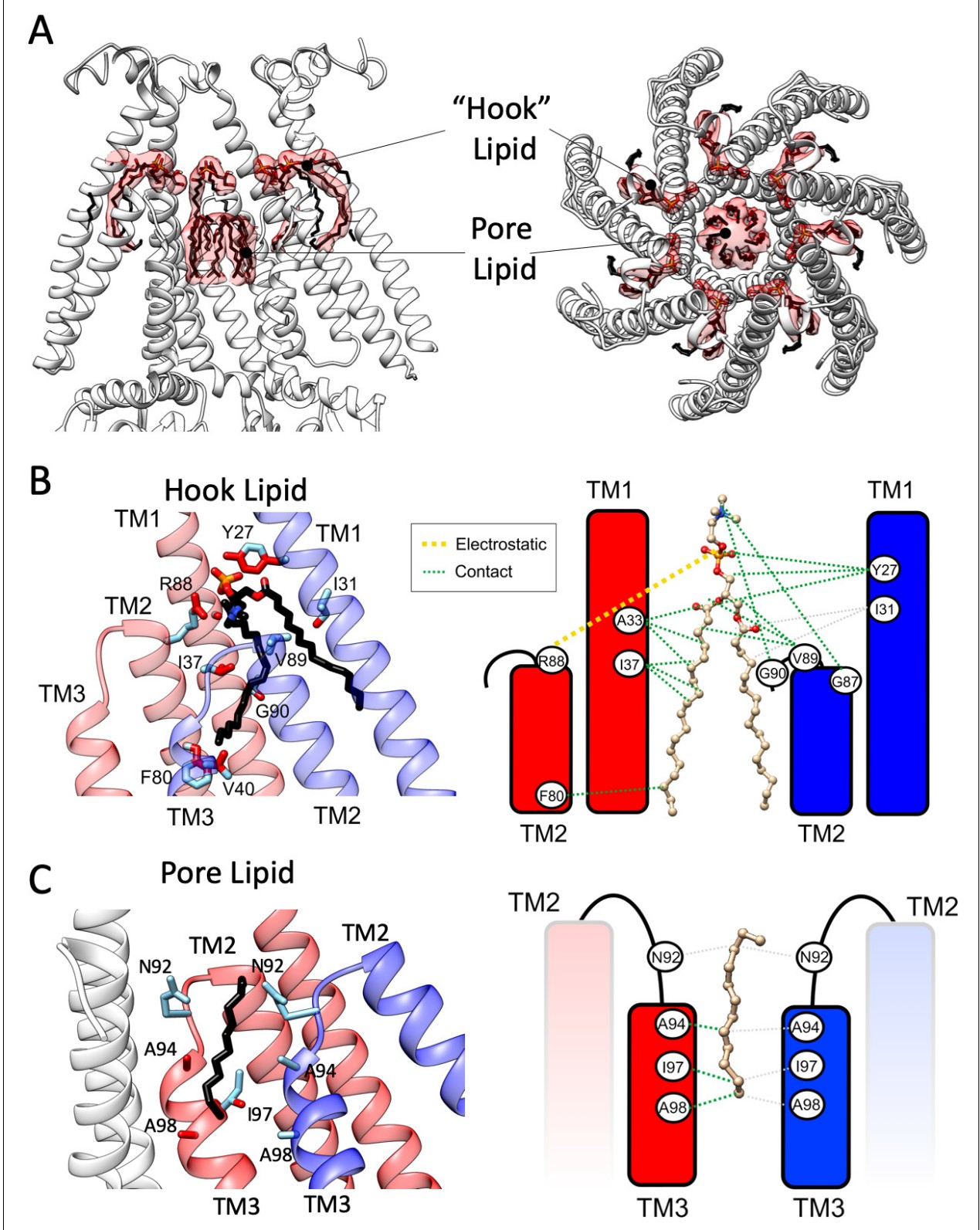

**Figure 5.** Bound lipid at the inner gate and the permeation pathway. (**A**) Side (left) and top (right) views of EM density (transparent red surface) associated with putative lipid molecules bound to MscS-ND (shown in white cartoon representation). A 'hook' phospholipid is cradled at the subunit interface atop the TM2-TM3 hairpin, while seven individual acyl chains line the permeation pathway along TM3, above the narrowest portion of the gate. (**B**) Contact map and coordination of the hook lipid. Left, TM helices from two adjacent subunits (red and blue) are shown. Key interactions are

*Figure 5 continued on next page*

*Figure 5 continued*

highlighted for R88 (in the red subunit) with the head group nitrogen (PC or PE) and Y27 (in the blue subunit) with the phosphate group. Residues within van der Waals distances are shown in red. Right, a cartoon representation of the contact/coordination map. (C) Same as (B), but with the pore lipid acyl chains.

The online version of this article includes the following figure supplement(s) for figure 5:

**Figure supplement 1.** Bound Lipids in MscS.
**Figure supplement 2.** Properties of the MscS Permeation Pathway.
**Figure supplement 3.** Bound Lipids are also found in Detergent-Based Structures.

corresponds to the location of the phosphate group in the hook lipid (*Figure 5—figure supplement 3B*), implying unresolved lipid-bound density in MscS crystals. However, the high resolution 5AJI expanded crystal structure does not show either lipids, which might be due to the lipid being solubilized away in the detergent environment when not protected by the closed conformational N-terminal loop. That MscS engages in close interactions with membrane lipids has in fact been suggested earlier based on non-denaturing mass spectroscopy (MS) (*Pliotas et al., 2015*). And while MS and chemical extraction helped identify at least five phospholipids per MscS (mostly PE) the predicted placement and interaction with MscS (*Pliotas et al., 2015*) appears to be incompatible with those observed in the present EM densities (*Figure 5*). Our EM density does not show explicit lipid density in the TM2-TM3 pockets. However, lipids bound to the TM pockets have been reported recently in ND-reconstituted MscS (*Rasmussen et al., 2019*), but whether or not these lipids play a functional role remains to be established.

Previous studies of closed state water permeation in MscS (*Anishkin and Sukharev, 2004*; *Spronk et al., 2006*) have shown that in spite of a wide (~7 Å) diameter at rest, the hydrophobic characteristics of the pore lead to a functional occlusion by a 'vapor lock' mechanism. Assuming that the pore lipids do occlude the permeation path in the closed state we set out to evaluate their influence on water dynamics along the permeation pathway. We consider three conditions: closed MscS pore with no associated lipids, with only the hook lipid, or with both hook and pore lipids (*Figure 6A*). As reported (*Anishkin and Sukharev, 2004*), the permeation path in MscS with no bound lipids fluctuates between a vapor locked state and a filled state where water is able to permeate. *Figure 6A* shows that during a 10 ns MD run the running averages for water permeation in the absence of bound lipid fluctuates around two water molecules in a $3 \times 2$ Å cylinder centered at L105. Inclusion of the hook lipid atop the TM2-TM3 hairpin reduces the running average about one water molecule at a time. However, the addition of both hook and pore lipids thoroughly eliminates any water permeability. This is illustrated from side views and cross sections of the pore in *Figure 6B*. The substantial effect of the pore lipid on water dynamics further suggests that under physiological conditions the pore lipid might be able to act as a low dielectric blocker, suggesting that the transition to the open conformation in MscS could be accompanied by a reduction in the occupancy of the pore lipid along the permeation pathway.

## Discussion

### On the mechanism of force-from lipid gating in MscS

The new MscS structural features, membrane footprint and bound lipids all have important mechanistic consequences regarding MscS force transduction. For one, TM3b can no longer be considered an interfacial helix (*Bavi et al., 2016b*) as it is located almost 15 Å away from the membrane/water interface (*Figures 1A*,*2B*, left). This is confirmed by our five structures presented here and elsewhere (*Rasmussen et al., 2019*). The proposed gating mechanisms where lipids act as ligands are based on the assumption that the TM2/TM3a cavity is open and accessible to the lipid bilayer (*Pliotas et al., 2015*) are unlikely due to the location of the TM2/TM3a cavity (or TM pocket) in relation to the membrane annulus around MscS (*Figure 2B*) seems incompatible with a proposed phospholipid exchange between bilayer and TM pockets. Accordingly, the suggestion that lateral tension would 'pull' on lipid acyl chains located in TM pockets appears unrealistic. Although our data only shows marginal additional density within the TM pockets, a recent structure of MscS in nanodiscs does find density that has been interpreted as lipidic (*Rasmussen et al., 2019*). Because of the

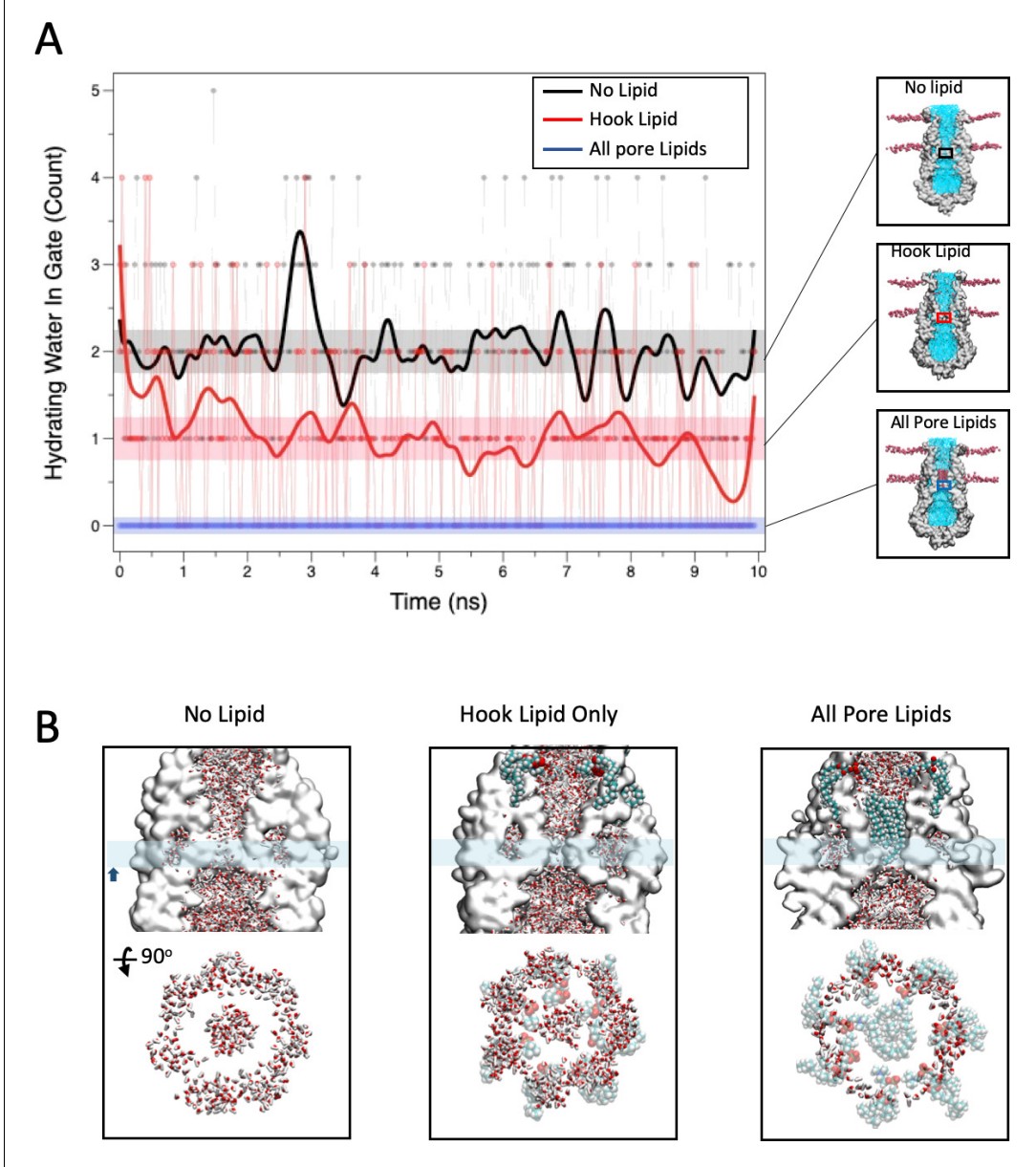

**Figure 6.** Role of bound lipids on the permeation pathway. (**A**) MD simulation of water permeation during 10 ns of equilibration under three lipid occupancy conditions: In the absence of bound lipids (black trace), with the hook lipid-bound (red trace) and with both hook and pore lipids bound (blue trace). Waters were counted in a 3 × 2 Å cylinder that includes the L105 residue (shown in the insets). (**B**) Close-up of the MscS permeation path under the three conditions described in (**A**). In each case the top image shows side views with water molecules in stick representation and the hook and bound lipids as VDW spheres. The cyan bar represents the slab thickness of the cross section in the bottom image (rotated 90 degrees).

volume change observed in the TM pockets of the expanded MscS conformation (*Lai et al., 2013*; *Pliotas et al., 2015*) it is expected that this lipid density should change during gating. The relative contribution of all bound lipids to force transduction in MscS will require further investigation.

Our structure in nanodiscs reveals much of the anchor domain locked in the closed state. We suggest that the anchor domain must play a significant role in the mechanism of mechanotransduction in MscS. This is based on three primary observations. First, deletion of the anchor domain renders MscS unresponsive to tension, although it does not preclude its folding and oligomeric assembly (*Figures 2A,4B*); second, limiting the conformational flexibility of the anchor domain by the N-terminal His-x6 'bundle' leads to non-functional MscS, and deleting the His-x6 'bundle' via proteolysis restores activity to WT level (*Figure 4—figure supplement 1B-D*); finally, we show that the anchor

domain is unusually sensitive to mutagenesis with up to 70% of its amino acids leading to LOF (or severe GOF) phenotypes upon mutation to ALA (*Figure 4B*). This is a much more severe effect than previous surface mutagenesis efforts on the basis of the 'incomplete' crystal structure. The significant functional and structural role of the anchor domain must now be considered in any force transduction model.

Earlier spectroscopic work (*Vasquez et al., 2008*) provides excellent clues regarding the types of conformational changes expected at the periplasmic side of the channel during its transition towards the open state. Changes in the NiEdda (water) accessibility between closed (at rest, in liposomes) and open MscS conformations (after LysoPC activation) show a massive reduction in water accessibility for the majority of the anchor domain (*Figure 7A*) when compared to the values at rest (*Figure 7—figure supplement 1*). This suggests that in the open state the anchor domain transitions to a deeper location in the outer leaflet of the bilayer (most likely as an extension of TM1), while it tilts and moves away from the permeation pathway (as seen in the MscS expanded conformations, *Lai et al., 2013*; *Wang et al., 2008*).

We find that at least two general gating models are consistent with both the data provided by the present MscS-ND structure and the conformational changes observed crystallographically (*Figure 7B*). In the first one (Open 1), the hook lipid remains bound to its pocket, allosterically coupling intramembrane forces with anchor domain rearrangements, TM1-TM2 hairpin reorientation and the expansion of the TM3a inner bundle. The second possibility (Open 2), would be reminiscent of the mechanism proposed to explain mechanosensitivity in the K2P channels TRAAK (*Brohawn, 2015*). As such, membrane stretch would trigger rearrangements at the anchor domain and TM1-TM2 hairpin, destabilizing the hook lipid pocket and leading to diffusion of the hook lipid into the bilayer. In turn, the release of the hook lipid triggers expansion of the TM3a inner bundle, opening the channel. Of course, in both gating models, channel opening ought to be accompanied by a release of the pore lipid, unblocking of the permeation path. Further research is needed to elucidate potential mechanisms of pore lipid dynamics.

The present results suggest that, for MscS, the energetic differences derived from tension changes in the plane of the lipid bilayer should be evaluated in the context of key lipids bound to mechanistically important regions of the channel (TM2 and TM3a). Functionally, MscS behaves as a lipo-protein complex, where the hook lipid may help transduce bilayer forces and the pore lipid is poised to influence ion and water fluxes at rest. These interactions, together with the revised location of the lipid-protein interface must be accounted for by mechanotransduction models which strong allosteric coupling between TM segments and the lipid-protein interface.

## Materials and methods

**Key resources table**

| Reagent type (species) or resource | Designation | Source or reference | Identifiers | Additional information |
|---|---|---|---|---|
| Gene (*E. coli* | | Addgene# 7855 | | 6x N-Terminal His-Tag |
| Gene (*E. coli*) | | Addgene# 20066 | | |
| *Strain, strain background (E. coli)* | MJF465 | Ian Booth and Samantha Miller, University of Aberdeen | | |
| Other | POPC | Avanti Polar Lipids | 850457C | |
| Other | POPG | Avanti Polar Lipids | 840457C | |
| Other | *E. coli* Polar Lipids | Avanti Polar Lipids | 100600C | |
| Other | Bio-Beads SM-2 Resin | Bio Rad | 1523920 | |

*Continued on next page*

*Continued*

| Reagent type (species) or resource | Designation | Source or reference | Identifiers | Additional information |
|---|---|---|---|---|
| Other | Quantifoil 2/2 Mesh 200 | Quantifoil | | |
| Other | Quantifoil 1.2/1.3 Mesh 300 | Quantifoil | | |
| Other | Octyl Maltoside, Fluorinated | Anatrace | O310F | |
| Other | n-Dodecyl-β-D-Maltopyranoside | Anatrace | D310A | |
| Other | Fos-Choline-14 | Anatrace | F312S | |
| *Strain, strain background (E. coli)* | Rosetta 2 | Millipore Sigma | 71400-3 | |
| Other | Thrombin | MP Biomedicals | 154163 | Bovine |

## MscS expression purification

Full-length *E. coli* MscS was expressed and purified as previously described (*Vásquez et al., 2007*). In brief, MscS was sub-cloned into pET28a containing a His$_6$ tag and a thrombin cleavage site on the N-termini. Rosetta 2 (Millipore Sigma) *E. coli* cells were transformed with MscS-pET28a vector and grown overnight in the presences of kanamycin and chloramphenicol. The cells were diluted 1:100 in LB medium and grown at 37°C to an OD$_{600}$ of 0.8-1.0. Before induction, the cell culture was supplemented to a final concentration of 0.4% glycerol and allowed to cool to 26°C, and protein expression was induced with 0.8mM IPTG. The cells were grown for 4h at 26°C and were harvested, and either were frozen at -80°C for later use or immediately resuspended in PBS pH 7.4 (Sigma), 10% glycerol, protease inhibitors, and homogenized (high-pressure homogenizer, EmulsiFlex-C3). The membranes were isolated via centrifugation at 100,000g for 30 min, and the pellet was resuspended in PBS and 10% glycerol. Solubilization was carried out in 1% Fos-Choline (Anatrace) 14 for 4-16h at 4°C. This resuspension was spun down at 100,000g for 30 min, and the supernatant supplemented with a final concentration of 5mM imidazole (Fisher) was incubated with cobalt resin(Clonetech) for 2-4h at 4 °C. The resin was washed with 20-bed volumes of 1 mM DDM(Anatrace), 10mM imidazole and 10% glycerol in PBS buffer. MscS was eluted in 1 mM DDM, 300mM imidazole, and 10% glycerol in PBS buffer. Unless explicitly stated MscS His, thrombin was added to cleave the his tag and incubated overnight. The final purification step was to run the protein on a Superdex 200 Increase 10/30 column (GE Healthcare) with 1 mM DDM and PBS buffer. The removal of glycerol is critical for EM grid preparation. The typical yield of MscS is about 5-8mg per liter of *E. coli*. For the MscS-Cryst construct, residues 2-26 residues were removed and subcloned into pQE70 and grown in MJF465 *E. coli* cells (to avoid co-assembly with chromosomal wt-MscS), a gift from Ian Booth (*Levina et al., 1999*). Typical yield of MscS-Cryst is less than 0.1mg per liter of MJF465 *E. coli*. Otherwise, the purification steps were the same. The MscS structure solved in DDM was solubilized in 1% DDM instead of Fos-Choline 14.

## MscS nanodisc preparation

MscS nanodiscs (ND) were prepared following previously described protocol (*Ritchie et al., 2009*). Several variants of ND scaffold proteins were tested, and Msp1 E3D1 was deemed the most homogenous by size exclusion. The molar ratio of MscS:MSP1 E3D1:Lipids was 7:10:650, respectively, after extensive optimizations. Each lipid solution of mixed micelles contained 30-50mM DDM with a final lipid concentration of 10-17mM. The compositions of the mixed micelles were either (1-palmitoyl-2-oleoyl-sn-glycero-3-phosphocholine) POPC and (1-palmitoyl-2-oleoylglycero-3-phosphoglycerol) POPG (4:1) or *E. coli* Polar Lipids (EPL). Nanodiscs were made by adding mix micelles to protein for 20 minutes on ice. MSP was added to the solution and incubated on ice for 5 minutes. The reconstitution mixture was incubated in activated bio beads (Biorad) overnight at 4°C. The detergent free mixture was run on a Superdex 200 Increase 10/30 column to separate the empty ND peak. The MscS ND peak was concentrated to ~2mg/ml and stored at 4°C.

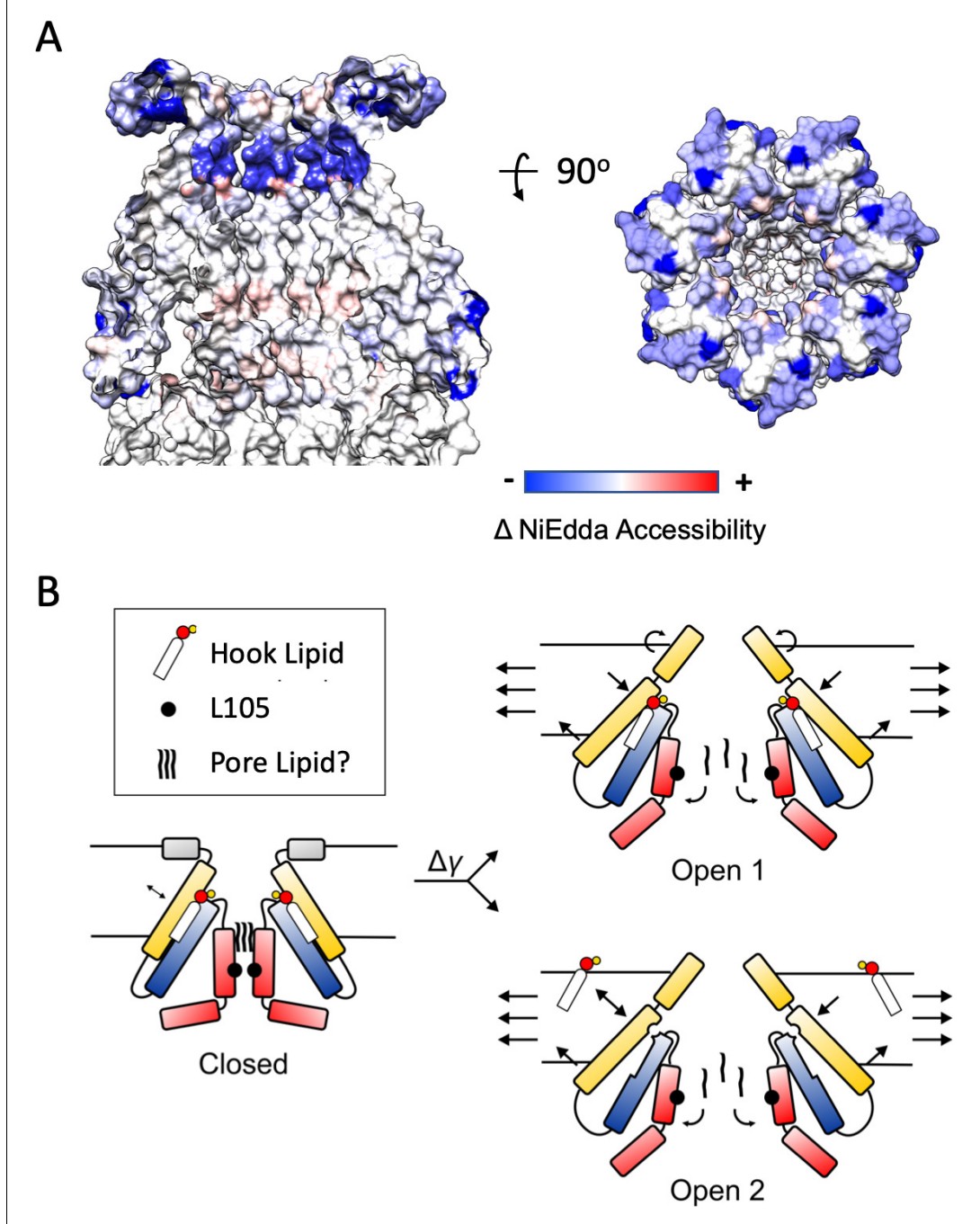

**Figure 7.** Mechanisms of Force-from-Lipid gating in MscS. (**A**) Extent and direction of environmental parameter changes upon MscS opening. Left, NiEdda accessibility (ΠNiEdda) mapped on MscS-ND transmembrane segments. Right, Changes in NiEdda accessibility (ΔΠNiEdda) mapped on MscS-ND transmembrane segments. Data from *Vasquez et al. (2008)*. Note, however that the EPR data were obtained on an unconstrained (though lipid embedded) N-terminal domain and should be treated as a general guide. A decrease in NiEdda accessibility corresponds to a decrease in accessibility to the aqueous milieu. (**B**) A lipid-centric mechanism of force transduction in MscS. In the nominal absence of external forces, MscS populates the resting, closed state represented by the MscS-ND structure (*Figure 1*). The non-conductive nature of the closed conformation is determined by a narrow cuff of hydrophobic residues around L105 (black circles) and above that, a plug of acyl chains from the pore lipid. The inter-subunit hook lipid links the top of the TM2-TM3a hairpin and the hydrophobic core of the bilayer. Applying forces in the plane of the bilayer lead to two gating scenarios, depending on whether the hook lipid stays bound to the open channel or not. In the first case (top) the expansion of TM1 in the periphery of MscS is allosterically communicated to the TM2-TM3 hairpin through the hook lipid, leading to the expansion of TM3a and release of the pore lipid. In the second option (bottom), expansion of TM1 destabilizes the binding of the hook lipid which exchanges with the bulk of the bilayer. Acting as a ligand, the release of the hook lipid triggers a conformational rearrangement in TM3a, with subsequent release of the pore lipid, leading to conduction.

*Figure 7 continued on next page*

*Figure 7 continued*

The online version of this article includes the following figure supplement(s) for figure 7:

**Figure supplement 1.** Mapping of EPR Data.

## EM data collection and structure determination

To help with orientation preferences and ice thickness, MscS ND was supplemented with Octyl Maltoside, Fluorinated (Anatrace) to a final concentration of 0.01%. MscS and was applied twice, with a filter paper blotting between each application, onto Mesh 200 2/1 or Mesh 300 1.2/1.3 Quantifoli holey carbon grids and flash frozen in a Vitrobot (Thermofisher) set at 3 seconds with a force of 3 with 100% humidity at 22°C. MscS His ND POPC:POPG, MscS No His ND POPC:POPG, and MscS DDM were collected on a Titan Krios with a K2 detector in counting mode with a GIF energy filter using Latitude S (Thermofisher). Movies were acquired at $1e^-/A^2$ per frame for 50 frames. MscS ND EPL was collected on Titan Krios with a Falcon 3 detector in counting mode. MscS-Cryst ND POPC:POPG was collected on Talos Artica with a Falcon 3 detector in counting mode. Movies were acquired at $1e^-/A^2$ per frame for 50 frames. Motion correction was performed using Motioncor2 (*Zheng et al., 2017*), and K2 movies were binned by 2. CTF estimation was done using CTFFIND4.1 (*Rohou and Grigorieff, 2015*). Initial particle picking was done using Eman's (*Tang et al., 2007*) neural net particle picker or Relion's built-in reference based auto picker and the coordinates were fed into Relion (*Scheres, 2012*) for particle extraction. Subsequent structure determination steps were done in Relion. An initial 2D refinement was done to remove non-particles and poor-quality classes, which were fed into 3D classification. 3D classification was performed using the MscS crystal structure as an initial model. After a subset of particles were identified for the final refinement, the particles underwent per particle CTF refinement followed by Bayesian polishing. The final 3D reconstruction used the classes with both top and side views and refined using a tight mask excluding the membrane and his-tag (when necessary) and C7 symmetry. Model building was based on the MscS crystal structure (PDBID: 2OAU) and used coot to build the remaining TM1, N-terminal domain, and the hook and pore lipids. EM density maps used in subsequent steps were not were not postprocessed or sharpened. While postprocessing and sharpening did improve the density for the most part, the N-terminal domain became much noiser. The initially built model was iteratively refined using Coot (*Emsley et al., 2010*), Chimera (*Pettersen et al., 2004*), MDFF (*McGreevy et al., 2014*) using VMD (*Humphrey et al., 1996*) and NAMD (*Phillips et al., 2005*) or ChimeraX (*Goddard et al., 2018*) with the ISOLDE (*Croll, 2018*) plugin, Arp/Warp (*Langer et al., 2008*), and Phenix's (*Adams et al., 2010*) real space refine.

## Downshock assay

Downshock assays were performed from a modified protocol from what was previously described (*Batiza et al., 2002*; *Vásquez et al., 2007*). MJF465 cells transformed with various MscS mutants in pEQ70 were grown modified Luria-Bertani (LB) medium with 500mM NaCl and 100μg/ml ampicillin (Fisher), 50 μg/ml kanamycin(Fisher), and 25 μg/ml chloramphenicol(Fisher) at 37°C to an $OD_{600}$ of 0.6. The cells were cooled to room temperature and induced with 1 mM IPTG (Fisher) for 2 hours at 25°C. The $OD_{600}$ was measured and downshocks were performed by diluting cells 1:50 into a modified LB medium at 50mM NaCl and 1:100 was plated on standard LB agar plates overnight at 37°C. The colonies on the LB agar plates were imaged and counted and normalized by the $OD_{600}$ readings. Additionally, to assess the expression of each mutant, a western blot was performed. The western blot of each MscS expressing mutant was from a pellet from the downshock experiment and resuspended in PBS and SDS to a final 1% solution. The lysate was then sonicated, ran on a 4-20% SDS-PAGE gel (Biorad), transferred to PVDF and probed with the Penta-HIS(Qiagen) primary and anti-mouse conjugated to Alexa 488 secondary.

## Phylogeny analyses

*Enterobacteriales* and *Vibrionales MscS* protein sequences were extracted from the complete proteomes in the NCBI Assembly database. From each proteome, only one protein showing the highest BLAST bit score (*Camacho et al., 2009*) to the *E. coli* MscS protein query was extracted. Sequences were aligned using MUSCLE (v.3.5) (*Edgar, 2004*), and the ML phylogeny was inferred using RAxML

(v.8.2.11) (*Stamatakis, 2014*) (best-fit model of evolution: LG+G+X). The schematic representation of the phylogeny was generated using iTOL (*Letunic and Bork, 2019*). The relative rate of evolution for each site was inferred from an alignment of *Enterobacteriales* MscS proteins using RAxML (v.8.2.11) (*Stamatakis, 2014*).The rate of evolution was mapped on protein structure using Chimera (*Pettersen et al., 2004*). The sequence logo was generated from an alignment of *Enterobacteriales* MscS proteins using WebLogo 3 (*Crooks et al., 2004*).

## Proteoliposome preparation and patch clamp electrophysiology

Proto-liposomes were prepared using Dehydration Rehydration (D/R) method as fully described in previous studies (*Nomura et al., 2015*). Briefly, Avanti soybean lipid dissolved in chloroform were dried with nitrogen flow to create a thin lipid film on a glass tube. The film was suspended and vortexed with D/R buffer (200 mM KCl, 5 mM HEPES, adjusted to pH 7.2 with KOH) and was subjected to 15 min of sonication. MscS was added to the lipid at a protein to lipid ratio of 1:200 (w/w) and incubated at 4°C for 1 h. to remove detergent, Biobeads (BioRad,Hercules,CA, USA) were added and incubated at 4°C overnight (minimum 3 h). The proteoliposomes were collected by ultracentrifugation and resuspended in 50 ml of D/R buffer. Small aliquots were spotted onto the glass cover slips and dehydrated overnight under vacuum conditions and at 4°C. The dried proteoliposomes were then rehydrated with 20-25 μl D/R buffer. After 6 h incubation at 4°C, they are ready for electrophysiological experimentation. The channel activity was examined in excised (inside-out) configuration. An isotonic recording solution were used in the bath and pipette (200 mM KCl, 40 mM MgCl2, and 5 mM HEPES adjusted to pH 7.2 with KOH). Borosilicate glass pipettes were pulled using Sutter micropipette puller (P-1000, Flaming/Brown). The resistance of the capillary pipettes was from 2 to 4 mOhm. The current was amplified with an Axopatch 200B amplifier (Molecular Devices, Sunnyvale, CA, USA), filtered at 2 kHz and the data acquired at 5 kHz with a Digidata 1322A (Axon instruments) interface using pCLAMP 10 acquisition software (Molecular Devices). Negative pressure was applied using High Speed Pressure Clamp-1 apparatus (ALA Scientific Instruments, Farmingdale, NY, USA).

## All-atom molecular dynamics (MD) simulation

CHARMM GUI was used to embed MscS structure into a POPC:POPG (4:1) bilayer mix (to mimic our nanodisc lipid composition) (*Jo et al., 2008*). Different computational models have been generated as listed in *Supplementary file 1* The equilibration steps were performed similarly to our previous MD simulation of EcMscL (*Bavi et al., 2016a*). TIP3P water molecule was used to solvate the system. The lipid and water molecules in close proximity to the channel (<0.5 Å and <0.5 Å respectively) were removed first. The system was ionized by 200 mM KCl. Short lipid tail randomization was done for 20 ps. Lipid and water were packed around the protein for 1 ns, while the Cα atoms in the protein were fixed. The restraint on the protein was released, and the equilibration run was performed for 60 ns. In order to simulate our system in an NPT ensemble, a modified Nosé-Hoover Langevin piston pressure control provided in NAMD (*Phillips et al., 2005*) was applied to control fluctuations in the barostat around the constant pressure of 1 atm, whereas the temperature was controlled at 298 K via Langevin dynamics. The Particle-Mesh Ewald (PME) method was used in all simulations to compute electrostatic interactions beyond a real-space cut-off of 1.2 nm using a Fourier grid spacing of 0.1 nm. van der Waals interactions were smoothly switched off at 8−10 Å. Periodic boundary conditions were applied in all three directions. The CHARMM c36 Force field was used for all MD calculations (*Brooks et al., 2009*). We used VMD and Chimera for visualization and illustration of our simulation results (*Humphrey et al., 1996*; *Pettersen et al., 2004*). We continued the equilibration until the RMSD values of the protein backbone over the equilibration time was plateaued (*Figure 2—figure supplement 2B*).

## Pore hydration calculations

Water molecules were counted for the last 10 ns of equilibration for each model. A custom TCL script was used (*Bavi, 2019* at https://github.com/Perozo-lab/PMF; copy archived at https://github.com/elifesciences-publications/PMF), which counts the number of hydrating water molecules that pass the central hydrophobic pore (i.e. L105) over the simulation period.

## Potential mean force (PMF) energy calculations

PMF calculations were performed using a similar approach to previous studies (*Corry and Thomas, 2012*; *Li et al., 2018*; *Shen and Guo, 2012*). Free-energy values for different bilayer-to-protein position along the Z-axis (bilayer thickness) (*Figure 2—figure supplement 2C*) were calculated by umbrella sampling simulations (*Egwolf and Roux, 2010*; *Torrie and Valleau, 1977*). Three different models were built to explore the most energetically favorable position of the lipid bilayer with respect to MscS along the Z-axis. The first model is Model 4 (*Supplementary file 1* ) where the MscS position in the bilayer has been determined based on the CRARMM-GUI potentials (*Jo et al., 2008*). We first defined our reaction coordinates as the distance between the Z coordinate of center mass of phosphate molecules of the lipid bilayer and Z coordinate of center mass of the pore-forming helices of MscS (i.e. TM3a, *Figure 2—figure supplement 2D*). The predicted model was shifted upward (in the +Z direction, i.e., along with the bilayer thickness) by ∼ 8 Å to match the bilayer position based on our Cryo-EM structure determined in nanodisc (Model 3, *Figure 4C*). The third model was built such that it was 16 Å higher than the position predicted by CHARMM-GUI (Model 5, *Figure 4C*). The starting configurations for the umbrella sampling simulations were taken from the MD trajectory of Model 3, Model 4 and Model 5 (varying in 1 Å steps from reaction coordinates of 16 Å to 31 Å) (*Figure 2—figure supplement 2B*).

As there is sufficient overlap between the sampling windows (*Figure 2—figure supplement 2D*), the number of windows is, therefore, enough to have an acceptable evaluation of the free energy landscape. A biasing harmonic potential force of 5 kcal/mol/Å$^2$ was used to constrain the position of the bilayer with respect to the protein. Therefore, 16 simulations were performed, where each simulation consisted of 1ns equilibration (no harmonic force) followed by 10 ns of production run (in the presence of harmonic force). Data were unbiased and combined using the weighed histogram analysis method using WHAM package (*Grossfield, 2010*). The minimum-energy path connecting the free-energy minima with respect to the reaction coordinate (bilayer-to-protein position) was shown in *Figure 4C*. We are aware there are differences between the ND membrane crossectional area (∼130Å diameter) vs. the membrane in MD (160Å x-y box), however we believe MscS is not restricted in either case as even at its widest predicted TM point, MscS is less than 75Å in diameter. Considering the crossectional area of a POPC lipid is ∼67Å$^2$ (*Bayburt and Sligar, 2010*) this maintains lipids several layers deep before hitting the edge (*Figure 4A*).

## Continuum Mean-Field calculations of the free energy change

Hydrophobic length of the protein, $d_p$, was determined based on the average Z distance between the center of mass (COM) of residue W16 to I48 (*Figure 2—figure supplement 3A*). The hydrophobic length of lipid, $d_l$, was calculated as the average Z distance between the COM of C1 atoms in the upper leaflet and the lower leaflet. Given the membrane thickness changes drastically from around the channel towards the boundary of our simulation box (*Figure 2—figure supplement 3B* left), we calculated the $d_l$ for lipids that are within r = 8Å of the protein (*Figure 2—figure supplement 3B* right). The hydrophobic mismatch length, $d_H$, then can be estimated as:

$$d_H = d_p - d_l \tag{1}$$

The maximum radius of curvature (*R*) and curvature (*C*) were estimated by measuring the *l* and *h* values as following (*Figure 2—figure supplement 3C*),

$$R = \frac{h}{2} + \frac{l^2}{8h} \quad \& \quad C = \frac{1}{R} \tag{2}$$

Here we investigate whether the binomial curve seen in our free energy diagram (*Figure 2—figure supplements 2,3D, E*), is due to change in the membrane curvature or due to hydrophobic mismatch between the protein and lipid bilayer at different Z distances (reaction coordinates). For this aim, we have monitored and measured the average curvature (*C*) and hydrophobic mismatch ($d_H$) values over the last 5 ns of each umbrella sampling window for the PMF calculation. The free energy contribution of curvature, $\Delta G_C$, and hydrophobic mismatch, $\Delta G_H$, can be described as the following phenomenological expressions (*Marsh, 2007*; *Kralj-Iglič et al., 1999*; *Kralj-Iglic et al., 1996*; *Bavi et al., 2016b*; *Svetina, 2015*).

$$\Delta G_C = \frac{1}{2} K_C \left( C_1 + C_2 - C_0 \right)^2 A + K_G C_1 C_2 A$$

$$\Delta G_H = \sqrt{2} \left( \frac{K_A^3 K_B}{t^6} \right)^{0.25} \frac{|d_p - d_l|^2}{4} 2\pi R_{ave} \tag{3}$$

Here for simplicity, it is assumed that since the MscS is an isotropic inclusion (i.e. cylindrical or conical in both principal planar directions), its insertion in the membrane causes symmetric curvature (i.e., $C_1 = C_2 = C$) in the membrane planar directions. Moreover, the intrinsic curvature of the protein has been assumed zero due to its shape. Where $K_A$, $K_C$ and $C_0$ are the area expansion and bending moduli and spontaneous curvature of the lipid bilayer which, based on previous and our current MD simulations are assumed to be 200 mN/m, 31 $k_B$T and ~0 Å⁻¹, respectively (*Akitake et al., 2005*; *Feller and Pastor, 1999*; *Marrink and Mark, 2001*). $C_0$ is assumed to be ~ 0 Å⁻¹ based on the flat shape of the bilayer (including the embedded MscS structure) at equilibrium (*Figure 6* and *Figure 2—figure supplement 3C*). $t$ is the global thickness and $A$ is the surface area of each monolayer and $R_{ave}$ is the average external radius of the transmembrane part of the protein which is assumed to be ~ 2 nm. $K_G$ is the elastic modulus for Gaussian curvature, which has experimentally approximated to be ~ - 0.1 $K_C$ (*Venable et al., 2015*; *Siegel and Kozlov, 2004*; *Templer et al., 1998*; *Marsh, 2007*; *Raghunathan et al., 2012*; *Venable et al., 2015*).

We also investigated whether variable $C$ as a function of lipid bilayer area would change the final values of the free energy. To do this, we discretized the membrane into concentric ribbons, $a_1$ to $a_n$ with $c_1$ to $c_n$ being their corresponding curvature values (*Figure 2—figure supplement 4*). Each value was averaged across different frames of the simulation (i.e. last 20 ns). Then the integral below (*Equation 4*) was used for calculating the free energy due to curvature change.

$$\Delta G_C = \frac{1}{2} K_C \int C^2 dA = \frac{1}{2} K_C \sum_{i=1}^{n} a_i c_i^2 \tag{4}$$

The resulting trend is similar to the case where we assumed a constant curvature across the bilayer, where the contribution of free energy due to change in the curvature is still an order of magnitude smaller than that of the hydrophobic mismatch (*Figure 2—figure supplement 3D-F*).

Software used in this project was curated by SBGrid (*Morin et al., 2013*).

## Acknowledgements

We thank Drs. Charles Cox, Boris Martinac, Ray Hulse, Yamuna Krishnan, Engin Ozkan, Minglei Zhao, Valeria Vasquez, Radomir Slovchov and the members of the Perozo lab for a healthy exchange of ideas and comments on the manuscript. Dr. Joseph W Thornton for fundamental insights on molecular phylogeny. Dr. Pedro Rodriguez and Wieslawa Milewski carried out the initial screen and mutant generation, respectively. We thank Ian Booth and Samatha Miller for their contribution of plasmids and bacterial strains. Michael Clark inspired the first sentence of this work. We are grateful to Drs. Lorenzo Pesce, Abhishek Singharoy and Rong Shen for invaluable assistance. This research was, in part, supported by the National Cancer Institute's National Cryo-EM Facility at the Frederick National Laboratory for Cancer Research under contract HSSN261200800001E. We thank Ulrich Baxa and Thomas J Edwards at NCEF for cryo-EM data collection, Pedro Rodriguez, Joe Austin II, and Tera Lavoie at the University of Chicago Advanced Electron Microscopy Facility for microscope maintenance and training. Dr. Mario Borgnia and Allen Hsu provided invaluable microscope time at the National Institute of Environmental Health Sciences. This work was supported by the Membrane Protein Structural Dynamics Consortium U54GM087519 and by NIGMS grant R01GM131191 to EP. AL was recipient of a BSCD Fellowship at the University of Chicago. NB is a Chicago Fellow. EM maps and atomic models have been deposited at the Electron Microscopy Data Bank (accession numbers EMD-20508, EMD-20510 and EMD-20509) and the Protein Data Back (entry codes 6PWN, 6PWP and 6PWO).

## Additional information

### Funding

| Funder | Grant reference number | Author |
|---|---|---|
| National Institute of General Medical Sciences | R01GM131191 | Eduardo Perozo |
| National Institute of General Medical Sciences | U54GM087519 | Eduardo Perozo |

The funders had no role in study design, data collection and interpretation, or the decision to submit the work for publication.

### Author contributions

Bharat Reddy, Conceptualization, Data curation, Formal analysis, Validation, Investigation, Visualization, Methodology; Navid Bavi, Allen Lu, Data curation, Formal analysis, Investigation; Yeonwoo Park, Formal analysis, Investigation; Eduardo Perozo, Conceptualization, Formal analysis, Supervision, Funding acquisition, Investigation, Visualization

### Author ORCIDs

Eduardo Perozo (iD) https://orcid.org/0000-0001-7132-2793

### Decision letter and Author response

Decision letter https://doi.org/10.7554/eLife.50486.sa1
Author response https://doi.org/10.7554/eLife.50486.sa2

# Additional files

### Supplementary files

- Supplementary file 1. Data Tables:Cryo-EM data collection and PMF parameters.

- Transparent reporting form

### Data availability

EM maps and atomic models have been deposited at the Electron Microscopy Data Bank (accession numbers EMD-20508, EMD-20510, EMD-20509, and EMD-20148) and the Protein Data Back (entry codes 6PWN, 6PWP and 6PWO).

The following datasets were generated:

| Author(s) | Year | Dataset title | Dataset URL | Database and Identifier |
|---|---|---|---|---|
| Reddy BG, Perozo E | 2019 | MscS Nanodisc with N-terminal His-Tag | https://www.rcsb.org/structure/6PWN | RCSB Protein Data Bank, 6PWN |
| Reddy BG, Perozo E | 2019 | MscS Nanodisc | https://www.rcsb.org/structure/6PWP | RCSB Protein Data Bank, 6PWP |
| Reddy BG, Perozo E | 2019 | MscS DDM | https://www.rcsb.org/structure/6PWO | RCSB Protein Data Bank, 6PWO |
| Reddy BG, Perozo E | 2019 | MscS Nanodisc with N-terminal His-Tag | https://www.ebi.ac.uk/pdbe/entry/emdb/EMD-20508 | Electron Microscopy Data Bank, EMD-20508 |
| Reddy BG, Perozo E | 2019 | MscS Nanodisc | https://www.ebi.ac.uk/pdbe/entry/emdb/EMD-20510 | Electron Microscopy Data Bank, EMD-20510 |
| Reddy BG, Perozo E | 2019 | MscS DDM | https://www.ebi.ac.uk/pdbe/entry/emdb/EMD-20509 | Electron Microscopy Data Bank, EMD-20509 |

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
