## [Decision Letter]

**Acceptance summary:**

The ion channel MscS has been an important model system to study the physical principles that explain molecular mechanosensation – which remain to be completely understood. In this article, Perozo and co-workers report single-particle cryo-EM structures of the MscS in lipid nanodiscs. These structures reveal the position of the channel relative to the membrane, which had been previously misjudged based on then-available structures. Specifically, the new data shows the transmembrane region is shifted by ~14 Å, taking TM3a and much of TM2 out of the membrane, and placing more of TM1 inside the membrane. These insights are consistent with free-energy calculations based on MD simulations. The structures also reveal the fold of a previously unrecognized 'anchor domain' at the N-terminus (of TM1), as well as ordered lipid molecules seemingly bound along the periphery of the channel as well as inside the pore. These are important insights that will no doubt serve as the foundation for future studies of the mechanism of gating of MscS and related mechanosensitive channels.

**Decision letter after peer review:**

Thank you for submitting your article "Molecular basis of force-from-lipidsgating in the mechanosensitive channel MscS" for consideration by *eLife*. Your article has been reviewed by three peer reviewers, and the evaluation has been overseen by José D. Faraldo-Gomez as the Reviewing Editor andRichard Aldrich as the Senior Editor. The following individuals involved in review of your submission have agreed to reveal their identity: Roderick MacKinnon (Reviewer #2); Sergei Sukharev (Reviewer #3).

The reviewers have discussed the reviews with one another and the Reviewing Editor has drafted this decision to help you prepare a revised submission.

The reviewers find that the central conclusions of the article are "well supported" and "convincing". By contrast, the computational examination of the mechanism of channel opening, and of the role that the bound lipids might play in this process, was perceived as too preliminary and inconclusive.

Essential revisions:

1) The reviewers agree that the computational section that examines the gating mechanism of the channel in response to a bilayer expansion and the potential role that bound lipids might have in this process is too preliminary and incomplete for publication in *eLife*. The reviewers are skeptical that these challenging questions can be addressed convincingly within the limited timeframe permitted to make revisions (2 months). Thus, we very strongly recommend that this element be removed from the article. The reviewers agree that this change will not diminish the value of the study, but rather the opposite. The authors are however encouraged to pursue this kind of studies in the future; some of the reviewers' comments are therefore provided below, as they might be useful to the authors:

– As the data stands, it is concerning that the process induced is clearly far from equilibrium, as a very large perturbation in membrane morphology is created in a very short time. To establish statistical significance on the basis of this kind of data, the authors ought to carry out and analyze a large number of repeats; alternatively, the authors could change their simulation protocol so that the system remains close to equilibrium throughout the perturbation of the membrane.

– The surface tension of the simulated membrane was increased from 50 mN/m at rest to 150 mN/m. The authors state that this is several times larger than the experimental value so that the opening can be observed in shorter times. But the lytic tension of ordinary lipid membranes is 10-15 mN/m, which means the tension in the simulation goes from about 5 times lytic to about 15 times lytic. What happens to the bilayer in the simulation? These simulated tensions deserve some careful thought, consideration and explanation for the reader.

– Forcing MscS opening by applying 150 mN/m of tension within 10 ns is a regime that imposes unnatural force and time scales on the transition. The bilayer gets much thinner than can be expected under MscS gating tension; hence the lipid-derived forces are acting at very different z levels. The bilayer thins so much that in videos it looks like the outer monolayer reaches the level of the "anchor" lipids that used to be close to the midplane. Also, why so much (50 mN/m) tension at rest? POPE seems to be pretty well parameterized in the latest release of CHARMM. Did the bilayer over-compact too much at zero tension or was the structure collapsing as in some previous simulations? This should be explained.

– The putative role of R88-associated lipids seems important and warrants a much more thorough MD study that would require several repetitions of channel opening under 'slow' regimes closer to equilibrium. Regarding ways to illustrate the conclusions, either cartoons should be presented (Figure 6B) or a real and convincing MD trajectory, not both. We suggest that in the next paper the authors devote special effort to the MD part which appears unfinished in the present form.

– The simulations do not seem to clearly reveal the role that either the 'hook lipids' or the 'pore lipids' might play in the mechanism of gating, or indeed whether they do play a role. For example, the authors propose that the pore lipids are a feature of the closed state, and that they must return the membrane upon channel opening. Simulations could in principle be used to evaluate the plausibility of this hypothesis, but the trajectories discussed in the manuscript are much too short to explore the dynamics of these lipids in a meaningful way. Similarly, much longer simulations (or simulations based on enhanced-sampling techniques) could be used to ascertain whether the sites occupied by lipid molecules in the cryo-EM structure indeed remain occupied in this closed state, as hypothesized, and to evaluate how this occupancy might depend on the membrane surface tension. In summary, this element of the study is promising but as it stands, it also appears inconclusive and not ready for publication.

2) The hypothesis that lipids are naturally present inside the pore in the resting state and move out during gating seems to be far-fetched. The lipid action in K2P channels, for instance, appears to be feasible due to clear fenestrations in the channel wall; similarly, a possible action of the 'hook' lipids here in MscS can be envisioned because there is an exchange path with the lipid bilayer. There is no obvious path that would connect the pore lipids with the bilayer. In this case, the detachment from the hydrophobic wall would imply a complete solvation by water, which would be energetically costly and slow. Because the headgroups of these lipids have not been resolved in any of the structures, would it be possible that these densities are fatty acids that came as products of lipid degradation? Fatty acid exchange is much more feasible due to higher solubility.

Given these concerns, we strongly recommend that the authors tone down their conclusions in regard to the observation of 'pore lipids' and that they discuss other plausible interpretations of the density signals. Otherwise, the authors are asked to provide an additional WT MscS structure obtained with a deliberate attempt to remove or minimize putative free-fatty acids, e.g. by incubating the nanodiscs with cyclodextrins or other type of fatty-acid absorbents. High-resolution structures might not be required to discern whether the densities inside the pore are still present or not.

[Editors' note: further revisions were requested prior to acceptance, as described below.]

Thank you for submitting the revised version of your article "Molecular basis of force-from-lipids gating in the mechanosensitive channel MscS" for consideration by *eLife*. Your article has been reviewed by two peer reviewers, and the evaluation has been overseen by José D. Faraldo-Gómez as the Reviewing Editor and Richard Aldrich as the Senior Editor. The following individuals involved in review of your resubmission have agreed to reveal their identity: Roderick MacKinnon (Reviewer #2); Sergei Sukharev (Reviewer #3).

The reviewers have discussed the reviews with one another and the Reviewing Editor has drafted this decision to help you prepare a revised submission.

Summary:

The reviewers believe the most important issues addressed in the earlier evaluation have been addressed, and that the revised manuscript will be worthy of publication in *eLife*, pending the following revisions:

The quantity defined as DGc in paragraph three of subsection “Continuum-Mean filed calculations of the free energy change” is not a free energy but a free-energy density, defined only locally at a point where the membrane curvature is C. The free energy that the authors seek should be an integral of this density over the area of membrane; the resulting values will be therefore considerably larger than the values currently plotted in Figure 3—figure supplement 3D. In addition, although it is valid as a first approximation to assume that at a given point (X, Y) the curvature C is same in the both directions, C(X,Y) cannot be assumed to be constant everywhere – i.e. the value of the abovementioned integral must be finite. Thus, the statement that "Although the contribution of curvature in our PMF calculations are minimal compared to the effect of hydrophobic mismatch" is not evident based on the data provided.

---

## [Author Response]

Essential revisions:1) The reviewers agree that the computational section that examines the gating mechanism of the channel in response to a bilayer expansion and the potential role that bound lipids might have in this process is too preliminary and incomplete for publication in eLife. The reviewers are skeptical that these challenging questions can be addressed convincingly within the limited timeframe permitted to make revisions (2 months). Thus, we very strongly recommend that this element be removed from the article. The reviewers agree that this change will not diminish the value of the study, but rather the opposite. The authors are however encouraged to pursue this kind of studies in the future; some of the reviewers' comments are therefore provided below, as they might be useful to the authors:

Indeed. We had been aware that a surface tension of 150 mN/m (applied solely to better sample any putative transitions in MscS) led to a membrane deformation that was perhaps too sharp and fast (see below). The intention was to include those bilayer expansion calculations as placeholders for more precise and ongoing calculations closer to equilibrium conditions (less intense surface tensions at 75, and 100 and 125 mN/m). These calculations (in replicates) started running before the manuscript was originally submitted. While the results are encouraging and reproducible, our intent is not to further delay publication of the present contribution. We agree with the reviewers that a dedicated computational study should be addressed in a separate publication. Therefore, we have removed this section of the manuscript.

– As the data stands, it is concerning that the process induced is clearly far from equilibrium, as a very large perturbation in membrane morphology is created in a very short time. To establish statistical significance on the basis of this kind of data, the authors ought to carry out and analyze a large number of repeats; alternatively, the authors could change their simulation protocol so that the system remains close to equilibrium throughout the perturbation of the membrane.

Agreed. These precise simulations (as mentioned above) are ongoing and shall be reported in a separate manuscript. Also, see below.

– The surface tension of the simulated membrane was increased from 50 mN/m at rest to 150 mN/m. The authors state that this is several times larger than the experimental value so that the opening can be observed in shorter times. But the lytic tension of ordinary lipid membranes is 10-15 mN/m, which means the tension in the simulation goes from about 5 times lytic to about 15 times lytic. What happens to the bilayer in the simulation? These simulated tensions deserve some careful thought, consideration and explanation for the reader.– Forcing MscS opening by applying 150 mN/m of tension within 10 ns is a regime that imposes unnatural force and time scales on the transition. The bilayer gets much thinner than can be expected under MscS gating tension; hence the lipid-derived forces are acting at very different z levels. The bilayer thins so much that in videos it looks like the outer monolayer reaches the level of the "anchor" lipids that used to be close to the midplane. Also, why so much (50 mN/m) tension at rest? POPE seems to be pretty well parameterized in the latest release of CHARMM. Did the bilayer over-compact too much at zero tension or was the structure collapsing as in some previous simulations? This should be explained.

These two comments touch on similar issues, so we combined our response here:

There might be a confusion between “surface tension” and “membrane tension”. The values we are referring to are of surface tension. We have included a figure (Author response image 1) from our previous publication to further clarify this point. Our original MD simulations used an NγP_z_T ensemble (1-3) in which “target surface tension” refers to the area of the pressure profile of the lipid bilayer near the lipid-water interface (Author response image 1). This is always a positive (attractive) tension of about 50 mN/m, which corresponds to the red area in the pressure profile below. In fact, without this level of surface tension the bilayer would otherwise shrink references [1,3,4].

The large surface tension values at the lipid-water interface are compensated to near zero by the compressive forces from entropic repulsion at 1), the core of the bilayer and 2) electrostatic repulsions between the headgroups (the blue areas in Author response image 1). Therefore, the total forces (membrane tension) along the Z-axis of the membranes at rest are indeed close to zero.

Membrane tension, as used in high speed patch clamp electrophysiology or micropipette aspiration rather refers to the latter case, i.e. total external forces that have been applied to the bilayer at rest, which cause an almost uniform increase in the lipid bilayer pressure profile from the equilibrium and subsequently an increase in the area per lipid. With this distinction, if the surface tension is set to i.e. 75 mN/m in an MD simulation, compared to equilibrium (~ 50 mN/m), it means that the total additional force (membrane tension) that has been applied to our MD system is ~ 25 mN/m (Author response image 1).

Ultimately, however, we fully agree with the reviewers that the force in our MD simulation is still several times higher than the experimental range (i.e., ~ 4 x when the surface tension is set to 75 mN/m). This is given that the midpoint of activation of MscS is ~ 5-6 mN/m (references 5-7) or as pointed out by the reviewers the lytic tension of lipid bilayers are usually <15 mN/m. Nonetheless, this is a common issue with these types of MD simulations and is a ‘necessary evil’ to be able to capture noticeable gating transitions within limited computational time scales (~200-300 ns).

**Author response image 1. respfig1:** Difference between the surface tension and overall membrane tension based on the lateral pressure profile of the bilayer. (**A**) The area shaded with red shows where regarded as “surface tension”, and blue the repulsive forces at the tail region of the bilayer and between the headgroups. (**B**) Illustration of how the pressure profile of the lipid bilayer in the presence of an MS channel (MscL) changes at different surface tensions of 50 mN/m and 75 mN/m in an NγPzT ensemble (Adapted from Bavi et al., 2016 (reference 4)). As shown, when the target surface tension is 50 mN/m the pressure profile of the lipid shows little or no change in the presence of the channels. Changes in the pressure profile only start to be noticed at 75 mN/m, particularly at the lipid water interface. In this condition (target surface tension of 75 mN/m; blue line), compared to a non-stressed bilayer (equilibrium; dark green) and based on the area under the pressure profile we estimate a mean membrane tension of 25.5 mN/m has been applied on the membrane, which is about 4x the experimental value required to activate MscS in patch clamp experiments.

– The putative role of R88-associated lipids seems important and warrants a much more thorough MD study that would require several repetitions of channel opening under 'slow' regimes closer to equilibrium. Regarding ways to illustrate the conclusions, either cartoons should be presented (Figure 6B) or a real and convincing MD trajectory, not both. We suggest that in the next paper the authors devote special effort to the MD part which appears unfinished in the present form.

Fully agree. Also, see response to comments above. We have eliminated the explicit trajectories and summarize our conclusion in cartoon form.

– The simulations do not seem to clearly reveal the role that either the 'hook lipids' or the 'pore lipids' might play in the mechanism of gating, or indeed whether they do play a role. For example, the authors propose that the pore lipids are a feature of the closed state, and that they must return the membrane upon channel opening. Simulations could in principle be used to evaluate the plausibility of this hypothesis, but the trajectories discussed in the manuscript are much too short to explore the dynamics of these lipids in a meaningful way. Similarly, much longer simulations (or simulations based on enhanced-sampling techniques) could be used to ascertain whether the sites occupied by lipid molecules in the cryo-EM structure indeed remain occupied in this closed state, as hypothesized, and to evaluate how this occupancy might depend on the membrane surface tension. In summary, this element of the study is promising but as it stands, it also appears inconclusive and not ready for publication.

We agree with the gist of these comments. Longer and targeted simulations will likely help discriminate between the different possible roles of the hook and pore lipids on MscS gating. As stated throughout this overall response to reviewers comments, the gating mechanism simulations are ongoing and shall be reported in a separate publication. Specifically, at identical external forces and computational conditions, we compared the channel opening with and without the presence of the hooked and pore lipids. We are also comparing all these scenarios with the model based on “previously believed” position of the bilayer systems with respect to the protein.

We do however believe that the simple evaluation of the closed state water occupancy (with and without pore lipids) and its influence on “vapor lock” mechanism are still informative and will be somewhat toned down but remain in the manuscript. Also, see response below.

2) The hypothesis that lipids are naturally present inside the pore in the resting state and move out during gating seems to be far-fetched. The lipid action in K2P channels, for instance, appears to be feasible due to clear fenestrations in the channel wall; similarly, a possible action of the 'hook' lipids here in MscS can be envisioned because there is an exchange path with the lipid bilayer. There is no obvious path that would connect the pore lipids with the bilayer. In this case, the detachment from the hydrophobic wall would imply a complete solvation by water, which would be energetically costly and slow. Because the headgroups of these lipids have not been resolved in any of the structures, would it be possible that these densities are fatty acids that came as products of lipid degradation? Fatty acid exchange is much more feasible due to higher solubility.Given these concerns, we strongly recommend that the authors tone down their conclusions in regard to the observation of 'pore lipids' and that they discuss other plausible interpretations of the density signals. Otherwise, the authors are asked to provide an additional WT MscS structure obtained with a deliberate attempt to remove or minimize putative free-fatty acids, e.g. by incubating the nanodiscs with cyclodextrins or other type of fatty-acid absorbents. High-resolution structures might not be required to discern whether the densities inside the pore are still present or not.

We have indeed toned down our interpretation and expanded our discussion regarding the identity of the “pore lipids”. The density we observe supports the likely presence of some type of fatty acid chains in the pore. Indeed, we were able to fit a hexadecamer fatty acid within the density. As understood by the reviewers, likely due to dynamics/resolution we were unable to resolve a putative head group. We see one of two possibilities of where the head group can come from. Either the fatty acid chain could potentially thread through a gap between G104 residues, where MOLEonline (https://mole.upol.cz) supports the presence of a pathway in our structure (Author response image 2); alternatively, the lipid could come from the monolayer on the periplasmic side. We stand by our finding: The density we associate to the “pore lipids” is a robust feature of the density signal, it is present in every single one of our high-res FC14-extracted ND structures (even in DDM) and is also prominently featured in the Rasmussen structure (using different biochemical approaches). Ultimately, whatever it this density represents (and we still hypothesize it is lipidic), we firmly argue that it is not a consequence of lipid “degradation” or poor biochemical handling of our sample. Further, we are not keen on carrying out the cyclodextrin experiment. Given the location of the pore lipids (closest extracellular access is through a ~23Å opening, ~30 Å from the aqueous solution, at its shortest) it is unlikely that any cyclodextrin would be able to effectively access and extract any of it. Thus, while we hypothesize that these putative pore lipids contribute to a more effective “vapor lock” in the closed state, there are still aspects of the gating mechanism that are both fascinating and mysterious. Along those lines, we have changed the manuscript accordingly: softening our conclusions in this regard while reporting on the presence of the pore lipid density right above the narrowest point of the MscS permeation pathway.

**Author response image 2. respfig2:** Predicted pathway connecting the location of the putative pore lipids in the permeation path with the intracellular TM2/TM3a cavity. Volume predicted by MOLEonline (https://mole.upol.cz) is depicted as a blue transparent envelope. The tan ribbon corresponds to the TM3 a and TM3b helices, with G104 colored red. An arrow points to the potential connecting gap between adjacent subunits.

[Editors' note: further revisions were requested prior to acceptance, as described below.]

The quantity defined as DGc in paragraph three of subsection “Continuum-Mean filed calculations of the free energy change” is not a free energy but a free-energy density, defined only locally at a point where the membrane curvature is C. The free energy that the authors seek should be an integral of this density over the area of membrane; the resulting values will be therefore considerably larger than the values currently plotted in Figure 3—figure supplement 3D. In addition, although it is valid as a first approximation to assume that at a given point (X, Y) the curvature C is same in the both directions, C(X,Y) cannot be assumed to be constant everywhere – i.e. the value of the abovementioned integral must be finite. Thus, the statement that "Although the contribution of curvature in our PMF calculations are minimal compared to the effect of hydrophobic mismatch" is not evident based on the data provided.

The reviewer is correct. We have now recalculated this free energy density as the integral of a series of concentric area rings (sed to reduce frame-to-frame-variability in the MD simulation run). We include that as a modified Figure 3—figure supplement 3 and a new Figure 3—figure supplement 4, with corresponding text and full methodological description in subsection “Continuum Mean-Field calculations of the free energy change”. The new calculations supports the original idea and demonstrates that “the contribution of free energy due to change in the curvature is still an order of magnitude smaller than that of the hydrophobic mismatch”.

References

[1] Feller, Scott E., and Richard W. Pastor. (1999) J Chem Phys 111: 1281-1287.

[2] Marrink, S. J., and A. E. Mark. (2001) J Physical Chem B105: 6122-6127.

[3] Gullingsrud, Justin, and Klaus Schulten. (2004) Biophys J 86: 3496-3509.

[4] Bavi, Navid, et al. (2016) Nature Comm 7 (2016): 11984.

[5] Akitake, Bradley, Andriy Anishkin, and Sergei Sukharev. (2005) JGP 125: 143-154.

[6] Nomura et al. (2012) PNAS 109 (2012): 8770-8775.

[7] Nomura et al. (2015) FASEB J 29: 4334-4345.